# Predicting response to enzalutamide and abiraterone in metastatic prostate cancer using whole-omics machine learning

Anouk C. de Jong [1,4], Alexandra Danyi [2,4], Job van Riet [1], Ronald de Wit[1], Martin Sjöström [3], Felix Feng [3], Jeroen de Ridder [2] & Martijn P. Lolkema [1] ✉

Response to androgen receptor signaling inhibitors (ARSI) varies widely in metastatic castration resistant prostate cancer (mCRPC). To improve treatment guidance, biomarkers are needed. We use whole-genomics (WGS; $n = 155$) with matching whole-transcriptomics (WTS; $n = 113$) from biopsies of ARSI-treated mCRPC patients for unbiased discovery of biomarkers and development of machine learning-based prediction models. Tumor mutational burden ($q < 0.001$), structural variants ($q < 0.05$), tandem duplications ($q < 0.05$) and deletions ($q < 0.05$) are enriched in poor responders, coupled with distinct transcriptomic expression profiles. Validating various classification models predicting treatment duration with ARSI on our internal and external mCRPC cohort reveals two best-performing models, based on the combination of prior treatment information with either the four combined enriched genomic markers or with overall transcriptomic profiles. In conclusion, predictive models combining genomic, transcriptomic, and clinical data can predict response to ARSI in mCRPC patients and, with additional optimization and prospective validation, could improve treatment guidance.

With approximately 350,000 men dying yearly of prostate cancer, prostate cancer is the fifth leading cause of cancer-related death worldwide[1]. Although early-phase prostate cancer is known for its favorable prognosis, the prognosis of metastatic prostate cancer is poor, especially when patients progress to the castration-resistant phase of the disease[2,3]. The treatment of metastatic castration-resistant prostate cancer (mCRPC) has significantly improved since the advent of second-generation androgen receptor signaling inhibitors (ARSI), like abiraterone acetate + prednisone (AAP) and enzalutamide[3–5]. However, response to these treatments varies widely between individual patients[4,5] To improve therapy guidance and optimize patient outcome, biomarkers, which can predict response before or soon after the start of therapy, are needed.

The existing biomarkers for treatment guidance in this setting are increasingly based on so-called liquid biopsies. It has been shown that five or more circulating tumor cells (CTCs) in 7.5 ml of blood and high levels of cell-free DNA (cfDNA) before the start of treatment are associated with a poor prognosis[6–8]. In addition, more detailed molecular analyses can be performed to predict resistance to ARSI. In CTCs, expression of androgen receptor variant 7 (AR-V7) is associated with resistance to ARSI, while this correlation is not found for chemotherapy[8–14]. To genotype cfDNA, gene panels targeting known driver and/or resistance-related genes are often used for sequencing or PCR. The most commonly identified alterations, that are associated with resistance to ARSI in patients, encompass AR mutations and amplifications[8,15–19]. Furthermore, *RB1* loss, *TP53* aberrations, *ZFHX3* deletions and *PI3K* pathway defects were associated with worse survival[8,15,19]. However, liquid biopsy-based analyses are mostly targeted to a certain set of genes and rely on patients having a high tumor-derived cfDNA fraction in the blood. Therefore, liquid biopsies

[1]Department of Medical Oncology, Erasmus MC Cancer Institute, Rotterdam, the Netherlands. [2]Center for Molecular Medicine, University Medical Center Utrecht, Utrecht, the Netherlands. [3]Department of Radiation Oncology, University of California, San Francisco, CA, USA. [4]These authors contributed equally: Anouk C. de Jong, Alexandra Danyi. ✉e-mail: m.lolkema@erasmusmc.nl

are less suitable for the discovery of potential biomarkers, that predict the outcome of treatment.

Whole genome and transcriptome sequencing (WGS and WTS) of tumor tissue provides extensive and detailed information about underlying genomic and functional aberrations of the malignancy. Rather than relying on a priori known targets using targeted gene-panels, studying the genome-wide somatic inventory may enable the unbiased discovery of biomarkers predicting treatment outcome. Prior work shows that genomic clusters could be linked to response to treatment, e.g., patients with microsatellite instability tend to respond well to immunotherapy and patients with a BRCA-like phenotype are likely to benefit from PARP-inhibition[20,21]. The value of sequencing the entire genome, including non-coding regions, for understanding tumor proliferation mechanisms is shown by the identification of an intergenic enhancer region upstream of *AR*, that is amplified in 81% of the mCRPC patients and correlates to increased *AR* expression[20,21]. Besides, WTS reveals that the Wnt/β-catenin pathway is enriched in enzalutamide-resistant patients in comparison to enzalutamide-naïve patients[22]. In addition, mutations in β-catenin and loss of 17q22 are solely found in enzalutamide-resistant patients and are associated with poor clinical outcome[22,23].

Statistical analysis of WGS and WTS data is challenging due to the extreme high number of features. In the last years, precision oncology often employs machine learning (ML) approaches to build predictive models in clinical and preclinical settings using genomic and transcriptomic information[24]. By analyzing the performance of ML models, the predictive power of these features can be assessed. Recently, advanced deep learning models have shown promise in the field. In 2018, a deep learning model effectively integrated multiple data modalities and leveraged large available training data (-1000 drug response experiments per compound)[25]. Within clinical patient cohorts, sample sizes are usually smaller, leading to inevitable over-parameterization and poor generalization performance when complex deep-learning models are used. Therefore, in such scenarios simple and strongly regularized ML models are preferred, as these generally suffer less from overfitting on limited training data and have already been proven to be efficient in similar contexts[26,27]. In addition, several methodological steps can be performed to handle small datasets. One such method is feature selection in which e.g., transcriptomic features, deemed as irrelevant for a particular response or genotype are removed[28]. Another widely applied procedure is dimensionality reduction in which small datasets with high feature dimensionality (e.g., transcriptome-wide expression) can be represented with reduced feature space and consequently be used in ML models with a lower risk of overfitting[29].

In this study, we aim to develop a ML-based classification model using WGS and WTS characteristics from biopsied metastatic malignancies to predict response of individual mCRPC patients to ARSI. To this end, we interrogate the full genomic inventory of metastatic malignancies from 155 mCRPC patients, who were subsequently treated with ARSI. In addition, matched WTS of these malignant tissues is available for 113 included mCRPC patients. Based on ARSI treatment duration, patients are categorized into good and poor responders. Subsequently, we determine and validate relevant clinical, genomic, and transcriptomic features for their usage as features within a ML-based approach to predict response to ARSI. Finally, we validate the performance of this classification model within an internal and external patient cohort.

## Results

### Included patients in discovery cohort (CPCT-02)
Between February 2015 and October 2019, 235 patients with mCRPC were included within CPCT-02 and treated with AAP or enzalutamide directly after a fresh-frozen biopsy[20,30]. Two patients were included twice, resulting in the inclusion of 233 unique patients. From 235

biopsies, 155 (66%) could be successfully analyzed by WGS. Eighty biopsies were excluded due to an unevaluable biopsy ($n = 42$), a biopsy of the primary tumor ($n = 13$), whole exome sequencing (WES) instead of WGS ($n = 11$), protocol violation ($n = 9$), missing treatment information ($n = 4$) and a second evaluable biopsy in combination with ARSI within one patient ($n = 1$). The second evaluable biopsy of this patient was excluded to prevent overfitting in the analyses. Matched WTS data of the malignant tissue was available for 113 patients (Fig. 1).

### Clinical characteristics and stratification of patients
Patients were stratified in good (≥180 days of treatment; $n = 66$), ambiguous (101–179 days of treatment; $n = 25$) and poor (≤100 days of treatment; $n = 64$) responders, based on treatment duration with ARSI (Fig. 1). Cut-off values were based on clinical practice (see Methods). Baseline characteristics for good, poor and ambiguous responders are shown in Table 1 and were compared for good and poor responders, as only these groups were included for biomarker discovery (see Methods). Good and poor responders were similar in age (mean ± SD: 68.1 ± 7.9 years and 69.5 ± 7.7 years, respectively, *adjusted p (q) = 1.000*). Poor responders showed a trend towards a higher proportion of biopsies, obtained from liver, compared to good responders (18.8% ($n = 12$) versus 3.0% ($n = 2$), *q = 0.076*). The proportion of patients treated with AAP and enzalutamide, respectively, after biopsy was comparable in the two groups (good responders 40.9% ($n = 27$) and 59.1% ($n = 39$), and poor responders 62.5% ($n = 40$) and 37.5% ($n = 24$), *q = 0.266*). Median treatment duration was 445 days ($Q_1$-$Q_3$: 242–NR) and 63 days ($Q_1$ - $Q_3$: 48–83) in the good and poor responder group, respectively. The number of prior systemic treatment lines was higher in poor responders than in good responders (median 2 treatment lines ($Q_1$ - $Q_3$: 1–3) vs 1 treatment line ($Q_1$ - $Q_3$: 0–1), $q < 0.001$). In detail, poor responders were more often previously treated with enzalutamide (37.5% ($n = 24$) vs 6.1% ($n = 4$), $q < 0.001$) than good responders. In addition, poor responding patients had a higher median PSA value at time of biopsy than good responding patients (140 ug/L ($Q_1$–$Q_3$: 58–390, $n = 35$) vs 23 ug/L ($Q_1$–$Q_3$: 13–92, $n = 44$), $q < 0.001$).

### Exploration of relevant WGS and WTS characteristics relating to response
To investigate relevant WGS and WTS features, relating to treatment response, and to design our subsequent classification models, we split our discovery cohort (CPCT-02) of matched WGS and WTS samples ($n = 113$) into a training ($n = 79$; 70%) and internal validation ($n = 34$; 30%) dataset. The training set contained a balanced number of good and poor responders, $n = 38$ and 41 respectively (Fig. 2).

### The genomic landscape of mCRPC patients, treated with ARSI
By utilizing WGS, we could inventory the genomic landscape of the good, ambiguous and poor responders ($n = 155$; Fig. 3).

Comparisons between good and poor responders were performed within the training set ($n = 79$). We observed significantly higher numbers of the tumor mutational burden (TMB; $q < 0.001$), total number of structural variants (SV; $q < 0.05$), total number of tandem duplications ($q < 0.05$) and total number of deletions ($q < 0.05$) within the poor responders compared to the good responders (Suppl. Figure 1a–d). In detail, the median (and Q1 - Q3) for poor responders vs. good responders of TMB, total SV, total number of tandem duplications and total numbers of deletions was observed to be 3.03 (2.32 − 4.3) vs. 2.21 (1.72 − 2.77), 349 (268 − 618) vs. 246 (172 − 377), 44.5 (24 − 80) vs. 30.5 (20 − 44) and 79.5 (59 − 116) vs. 64.5 (43 − 90), respectively.

We next assessed the mutational incidence of known genomic aberrations, related to ARSI resistance, which included genomic aberrations within *AR*, *TP53*, *PTEN*, *RB1*, *CTNNB1* and chromosomal arms aneuploidies (Suppl. Fig. 1e, f)[8,15–19]. No statistically significant differences between good and poor responders for

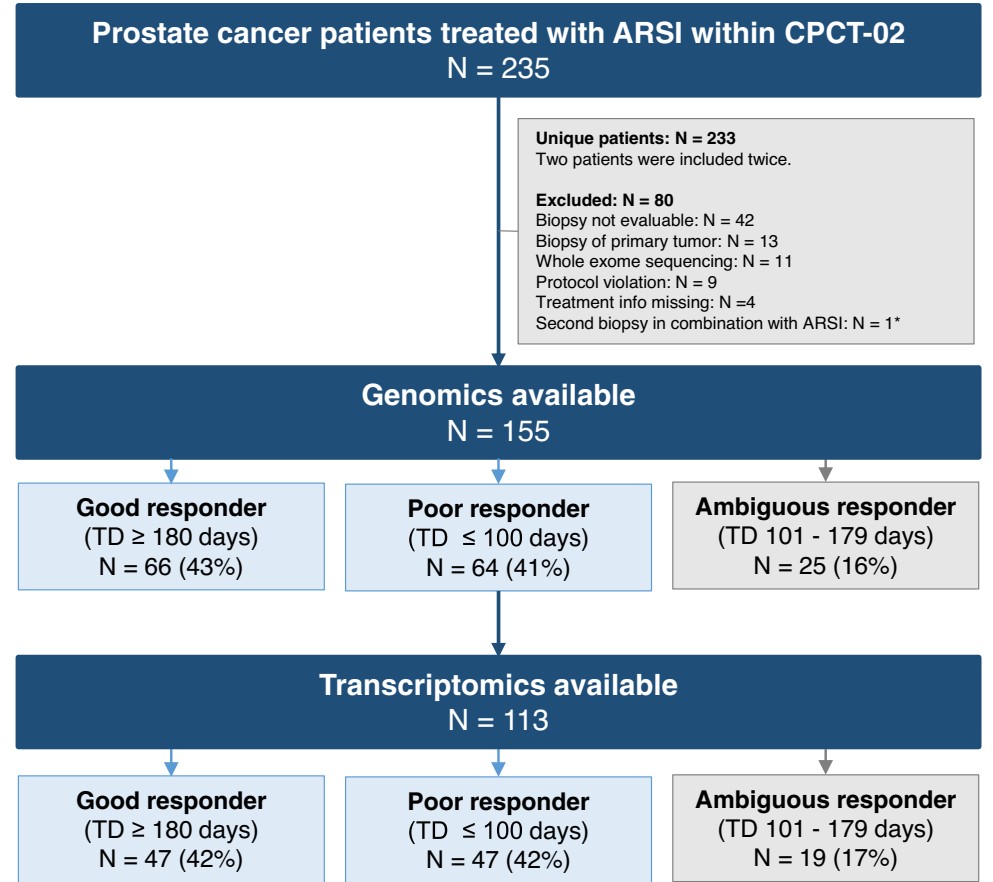

**Fig. 1 | Overview of mCRPC patients, treated with ARSI post-biopsy, within CPCT-02 cohort.** Of the 235 mCRPC patients, who were treated with androgen receptor signaling inhibitors (abiraterone acetate + prednisone (AAP) or enzalutamide; ARSI) and included in the CPCT-02 study, genomics and transcriptomics were available for 155 and 113 patients, respectively. Patients were stratified according to treatment duration with ARSI (TD). Only good and poor responders were used for the exploration of whole-omics data to prevent bias by the indistinct response of ambiguous responders. Ambiguous responders were included in the internal validation. One patient was included twice. To prevent overfitting, only the first biopsy was used.

these markers were found in the training set. In addition, we did not observe significant mutual-exclusivity for driver-genes, as detected by unbiased driver detection (dN/dS and/or GISTIC2). Genes with protein-coding aberrations within ≥20% of all samples within either the poor or good responder group were explored as well. No significant differences between good and poor responders were observed.

## Differentially expressed genes in good and poor responders to ARSI

Using the matched WTS from the training cohort ($n = 79$), we investigated differentially expressed genes (DEGs) between the good and poor responders for all protein-coding genes, which were not designated as putative biopsy-site specific markers (see Methods; Fig. 4). Using stringent criteria to select for uniform disparity between the two groups, we designated 151 genes as DEGs between the two groups (Fig. 4, Suppl. Table 1). In addition, we performed Gene-set Enrichment Analysis (GSEA) between the responder classes (Suppl. Figure 2a) and assessed the expression of AR-V7 (Suppl. Figure 2b).

Within the DEGs ($n = 151$), we observed uniform presence of genes regulating or attributed to epithelial-mesenchymal transition (EMT) such as *TIMP-3* and *TGFBI*, or found genes previously attributed to tumorigenesis, poor survival and/or aggressiveness such as *RGS2* and *SLC7A5 to* harbor higher expression within the poor responders vs. the good responders[31,32]. Conversely, genes attributed to the suppression of tumor growth and/or metastatic potential such as *RBM47* and *ENDOD1* were expressed in fewer quantity[33,34] (Suppl. Fig. 2a). We

did not observe dissimilar expression of AR-V7 between responder classes (Suppl. Fig. 2b).

Concordant, the GSEA revealed enriched expression of mechanisms and signaling, commonly reported within more aggressive forms of prostate cancer, including EMT, coupled with enriched inflammatory responses, TGF-β receptor signaling and TNFα signaling via NF-kB. In addition, the good responders revealed enrichment of the androgen response gene-set.

## Robustness assessment of differential expression analysis

After the initial differential gene expression analysis, we performed an out-of-sample analysis in a Leave-One-Out Cross-Validation (LOOCV) scheme to test the robustness of the selected DEGs due to our limited sample size (see Methods). As we observed notable variation within several DEGs between LOOCV folds, we suspected that a straightforward DESeq2-approach might possibly not provide robust results for classification purposes (Fig. 4). Therefore, we opted for an alternative methodology (Independent Component Analysis, described below) for feature selection during classification model development.

## Design of a machine learning-based classification model to predict response to ARSI

Our approach to generate classification models included three stages. First, we assessed relevant and robust WGS, WTS and clinical characteristics for treatment response using a LOOCV on the training set of samples with matched WTS and WGS ($n = 79$). Resulting models were subsequently compared and evaluated in the internal validation sets

**Table 1 | Baseline characteristics** Baseline table of clinical characteristics of the CPCT-02 cohort (DR-071; December 19th 2021)

| Characteristic | Responder category | | | Adj. *P* value* |
|---|---|---|---|---|
| | Good responder (TD ARSI ≥ 180 days, *n* = 66) | Poor responder (TD ARSI ≤ 100 days, *n* = 64) | Ambiguous responder (TD ARSI 101-179 days, *n* = 25) | Good vs poor responder** |
| **Age at biopsy in years** (mean ± SD) | 68.1 ± 7.9 | 69.5 ± 7.7 | 66.8 ± 8.6 | 1.000[a] |
| **Biopsy site** | | | | |
| Lymph node | 32 (48.5%) | 22 (34.4%) | 20 (80.0%) | >0.95[b] |
| Bone | 28 (42.4%) | 18 (28.1%) | 2 (8.0%) | >0.95[b] |
| Liver | 2 (3.0%) | 12 (18.8%) | 3 (12.0%) | 0.076[b] |
| Lung | 1 (1.5%) | 1 (1.6%) | 0 (0.0%) | >0.95[b] |
| Soft tissue | 3 (4.5%) | 6 (9.4%) | 0 (0.0%) | >0.95[b] |
| Other | 0 (0.0%) | 5 (7.8%) | 0 (0.0%) | NA[2] |
| **Started ARSI after biopsy** | | | | |
| AAP | 27 (40.9%) | 40 (62.5%) | 12 (48.0%) | 0.266[b] |
| Enzalutamide | 39 (59.1%) | 24 (37.5%) | 13 (52.0%) | |
| **Time between biopsy and start treatment in days** (median (IQR)) | 5.5 (0–14) | 5.0 (0–12) | 1.0 (0–7) | 1.000[c] |
| **Treatment duration in days** (median (IQR)) | 445 (242–NR) | 63 (48–83) | 133 (112–153) | NA |
| **Number of prior treatment lines** (median (IQR)) | 1 (0–1) | 2 (1–3) | 2 (0–3) | <0.001[c] |
| 0 | 25 (37.9%) | 7 (10.9%) | 7 (28.0%) | NA |
| 1 | 26 (39.4%) | 18 (39.1%) | 3 (12.0%) | NA |
| 2 | 8 (12.1%) | 21 (32.8%) | 4 (16.0%) | NA |
| 3+ | 7 (10.6%) | 18 (28.1%) | 11 (44.0%) | NA |
| **Prior therapies** | | | | |
| AAP | 4 (6.1%) | 10 (15.6%) | 6 (24.0%) | 1.000[b] |
| Enzalutamide | 4 (6.1%) | 24 (37.5%) | 10 (40.0%) | <0.001[b] |
| Other ARSI | 1 (1.5%) | 0 (0.0%) | 1 (4.0%) | 1.000[b] |
| Docetaxel | 35 (53.0%) | 50 (78.1%) | 15 (60%) | 1.000[b] |
| Cabazitaxel | 11 (16.7%) | 22 (34.4%) | 11 (44.0%) | 0.513[b] |
| Other chemotherapy | 0 (0.0%) | 4 (6.3%) | 0 (0.0%) | 1.000[b] |
| Radium-223 | 6 (9.1%) | 9 (14.1%) | 2 (8.0%) | 1.000[b] |
| Other treatment | 3 (4.5%) | 8 (12.5%) | 3 (12.0%) | 1.000[b] |
| **PSA at biopsy** (ug/L) (median (IQR)) | 23 (13–92) (*n* = 44) | 140 (58–390) (*n* = 35) | 13 (10–171) (*n* = 9) | <0.001[c] |

*P values were adjusted for multiple testing using Bonferroni (*n* = 19 tests).

**Two-sided statistical tests were performed to compare good and poor responders, as these patients were also used for further analysis and training of the classification model.

[a]Independent T-test.

[b]Fisher's Exact test + comparison of column proportions with z-test (*p* values adjusted according to Bonferroni).

[c]Mann–Whitney U test. The exact adjusted *p* value for number of prior treatment lines, prior enzalutamide and PSA at biopsy is 0.000076, 0.000209, and 0.00076, respectively. *TD* treatment duration, *ARSI* androgen receptor signaling inhibitor, *SD* standard deviation, *IQR* inter-quartile range, *NR* not reached, *NA* not applicable, *PSA* prostate specific antigen.

and finally, validated in an external cohort (see Methods; Fig. 2). In total, we utilized four schemes of classification models: 1) WGS-only, 2) WTS-only, 3) combined WGS and WTS and, 4) combined WGS and clinical co-variates. For these models, the remaining samples, that were not used for internal training (*n* = 79), were used for the internal validation. For the WGS-only and combined WGS/clinical variables models, this included 76 patients, spanning 28 good, 23 poor and 25 ambiguous responders. For the WTS-only and combined WGS/WGS models, this included 34 patients, spanning 9 good, 6 poor and 19 ambiguous responders. In addition, similar mCRPC patients from the West Coast Dream Team (WCDT) cohort, who were treated with ARSI as next therapy after biopsy, were used as external validation cohort[35]. Within this cohort, relevant WGS and WTS characteristics were available for 56 and 77 patients, respectively.

### Initial model performance assessment with LOOCV
**WGS-only classification model.** We utilized the four previously observed WGS characteristics, which revealed statistically significant differences between good and poor responders within the internal training set (TMB and the numbers of total structural variants, tandem

duplications and deletions) to train a Logistic Regression classifier. Performance was subsequently measured as Area Under the Curve (AUC) of 0.76, with a specificity and sensitivity of 49% and 79%, respectively (Fig. 5a and Table 2). Classifier hyperparameters were further tested in grid search, but no unequivocally better setting was found, when evaluating the model in LOOCV.

**WTS-only classification model.** Prior to WTS-based classification, dimensionality reduction was performed on the full transcriptome using multiple approaches (see Methods). Independent Component Analysis, sparse PCA, and conventional PCA were applied to the data with components ranging from 10 to 50, and consequently used as input in the training of linear Support Vector Classifier (SVC) models. The best overall performance was achieved with 40 independent components (ICs), with an AUC of 0.76, specificity of 83%, and sensitivity of 58% (Fig. 5a and Table 2).

### Combining WGS and WTS in ensemble classification models
Notable overlaps could be identified in the predicted true positives (true good responders, *n* = 18) and predicted true negatives (true poor

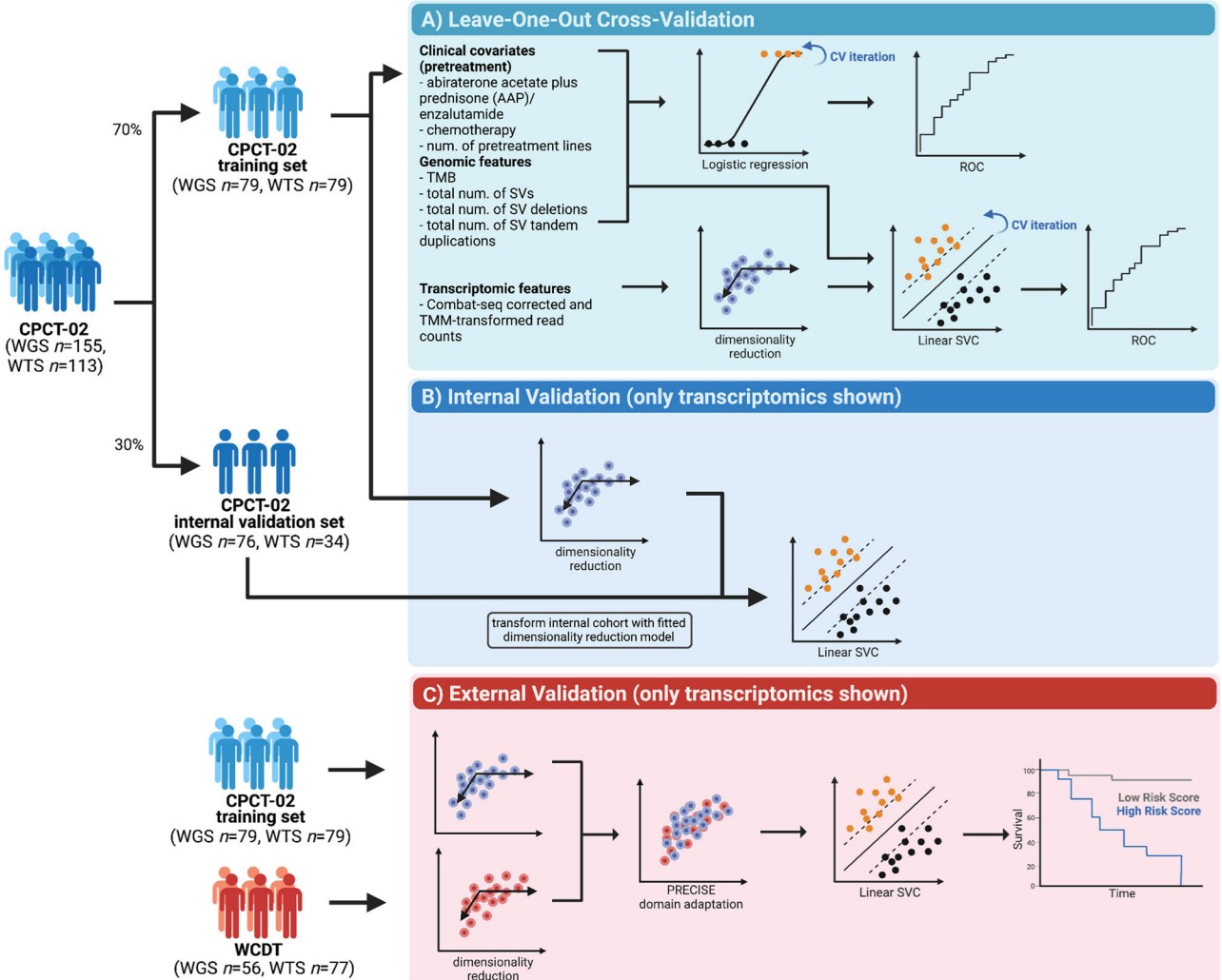

**Fig. 2 | Workflow of prediction model design and validation.** Machine learning pipeline design steps, using the CPCT-02 cohort (matched WGS/WTS n = 113) divided into a training set (n = 79) and an internal validation set (n = 34). External validation was performed on the WCDT cohort (WGS n = 56, WTS n = 77). **a** Steps in initial LOOCV performance assessment: Transcriptomics data was decomposed with Independent Component Analysis and the reduced feature space was used in linear support vector classifier (linear SVC). All classifiers that use transcriptomics data were linear SVC models. Statistically significant genomics features (TMB**, total number of tandem duplications*, total number of SVs*, total number of SV deletions*) and prior treatment features (ARSI, taxane-based chemotherapy, number of treatment lines) were tested separately in logistic regression classifiers. The combined WGS + WTS + ARSI data was tested in a linear SVC. Performance of the best classifiers (transcriptomics, "clinicogenomics" - genomics and prior treatment, "clinicotranscriptomics" - transcriptomics and prior treatment, WGS + WTS + prior treatment, genomics) was evaluated and visualized on ROC curves. **b** Internal validation of the best transcriptomics model (40 independent

components). The model was retrained on the full training set using the same features as in LOOCV (transcriptomics data decomposed with Independent Component Analysis). Then the transcriptomics data of the internal validation cohort was transformed with the 40 independent components model (which was previously fitted on the full training set), and predictions were made with the best transcriptomics model. (Other validated models were also retrained on the full training cohort, but no further pipeline adjustment was made - therefore not shown). **c** Transcriptomics data of the external validation cohort was decomposed with Independent Component Analysis into 40 components (independent of the training set). Then domain adaptation was performed on the decomposed transcriptomics data of the training set and the external cohort using PRECISE. Domain adapted training data matrix was used to train a Linear SVC on the transcriptomics-only, clinicotranscriptomics and WGS + WTS + prior treatment data. Performance of the classifier was evaluated on the domain adapted external data. (Other models did not require further pipeline adjustments - therefore not shown).

responders, n = 17) of the WGS-only and WTS-only models (Suppl. Fig. 3). The WGS-only model yielded better classification of good responders than the WTS-only model (79% vs. 58% sensitivity), whilst the WTS-only model yielded better classifications of poor responders (83% vs 49% specificity; Table 2). To investigate whether leveraging both WTS and WGS features would improve performance, we combined our best-performing WGS-only and WTS-only classification models using two ensembling approaches (see Methods). The stacking classifier resulted in an AUC of 0.76 (71% specificity / 71% sensitivity), whilst ensemble averaging resulted in an AUC of 0.81 (73% specificity /

68% sensitivity) (Fig. 5b and Table 2). The four WGS features and the WTS features from the best performing model (40 ICs) were also combined in two additional ensembling experiments (see Methods). The bagging classifier yielded an AUC of 0.76 (66% specificity / 71% sensitivity), while the multi-model averaging ensemble resulted in an AUC of 0.75 (59% specificity / 66% sensitivity) (Fig. 5b). Thus, the ensemble model that outperformed the WGS-only and WTS-only classification models was the averaging ensemble, which yielded an AUC of 0.81 compared to the WGS-only and WTS-only model with both an AUC of 0.76 (Fig. 5e).

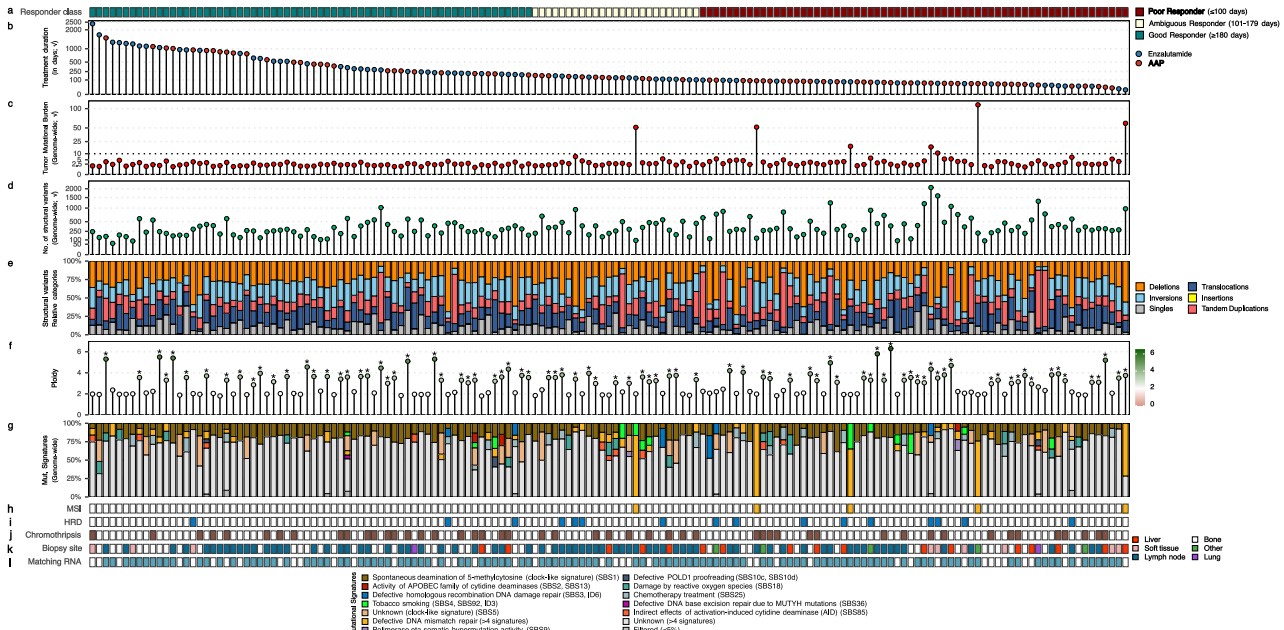

**Fig. 3 | Genomic landscape of the discovery cohort, ordered by ARSI treatment-duration and response category.** Overview of genome-wide characteristics of the discovery cohort (CPCT-02; DR-071; $n = 155$) ordered by descending treatment duration (abiraterone acetate + prednisone or enzalutamide). For each mCRPC patient, the following tracks are shown: **a** Responder category. The responder category of each sample, based on treatment duration (in days) on ARSI. **b** Type and duration of ARSI treatment. Y-axis representing the ARSI treatment duration (in days), whilst the coloring represents the type of given treatment (abiraterone acetate + prednisone in blue, enzalutamide in orange). Y-axis is shown in square-root transformed scale. **c** Tumor mutational burden (TMB). The number of genomic mutations averaged per megabase over the entire genome (TMB). Threshold for high-TMB status (TMB ≥ 10) is shown by a horizontal red dotted line. Y-axis is shown in square-root transformed scale. **d** Total no. of structural variants. The total number of structural variants (green) over the entire genome. This includes deletions, tandem duplications, translocations, inversions, insertions, and single-end breakpoints as detected by GRIDSS. Y-axis is shown in square-root transformed

scale. **e** Relative frequency of structural variant classes. Relative frequency of each of the structural variant categories; deletions in orange, inversions in light-blue, tandem duplications in red, translocations in dark-blue, insertions in yellow and single-end breakpoints (Singles) in grey. **f** Mean genome-wide ploidy. Mean genome-wide ploidy ranges from 0 (red) to 6 (green; hexaploidy). Common diploid status is shown in white. Samples which have undergone a whole-genome duplication (WGD) have been marked by an asterisk (*). **g** Relative frequency of COSMIC (v3.2) signatures, grouped per proposed etiology. The relative contribution of the COSMIC single-base substitution mutational signatures (v3) grouped and aggregated per proposed etiology. Per sample, signatures with <5 percent relative contribution were categorized under the "Filtered (<5%)" category. The proposed etiology of the signatures is denoted below. **h** Microsatellite-instability (MSI) status. **i** Homologous recombination deficiency (HRD) status as detected by CHORD. **j** Presence of chromothripsis. **k** Generalized biopsy location. l)Availability of matched whole-transcriptome sequencing.

## Addition of clinical data to the WGS- and WTS-based classification models

Compared to true good responders, true poor responders received more prior treatment lines for metastatic prostate cancer, including more frequently prior enzalutamide (Table 1). Therefore, we determined whether including information on whether patients had received prior treatment with ARSI and/or taxane-based chemotherapy and the number of respective treatment lines would increase the performance of the best-performing classification models. A classification model based solely on these clinical variables yielded an overall mediocre performance with a maximal AUC of 0.61 and sensitivity and specificity of 45% and 51%, respectively (Fig. 5c, Suppl. Fig. 4). However, we investigated whether a potential synergistic effect could be found by integrating a mixture of these clinical variables with the WGS and WTS data. Out of the models, combining WGS with clinical variables, the addition of prior/no prior ARSI as feature into the WGS-only model resulted in the highest performance increase compared to WGS-only. This combined 'clinicogenomics' model yielded an AUC of 0.81 with 66% specificity and 76% sensitivity (Fig. 5c, e and Table 2). The model that used WTS combined with prior/no prior ARSI also performed well, yielding an AUC of 0.82 with 73% specificity and 74% sensitivity (Fig. 5c, e and Table 2). A final combined model, which included both WTS and WGS features with prior/no prior ARSI, resulted in an AUC of 0.84 with 73% specificity and 74% sensitivity (Fig. 5c, e and Table 2).

## Shuffled label experiments

To confirm whether the presented classification models operate on meaningful underlying structures, random label permutation experiments were performed on the best models in LOOCV setting. The shuffled label experiments resulted in a median AUC of 0.50–0.51 for all models, with upper quartiles of the shuffled label experiments well below the AUC obtained using correctly labeled data (Fig. 5d). Based on these results, we concluded that our presented classification models indeed capture underlying patterns relating to the treatment response.

## Validation of final classification models

We validated our best-performing models in an internal and external validation cohort. Here, we describe the validation of one of the best performing models, which utilizes the four significant genomic characteristics and prior treatment with ARSI (clinicogenomics model), in detail (Table 2, Figs. 6 and 7). The other models were also successfully validated and the corresponding results are summarized in Table 2 and Suppl. Figs. 6 and 8.

## Internal validation cohort

For internal validation, we used 76 WGS samples of the CPCT-02 cohort, that were not used during training. This internal validation cohort encompassed 28 good, 23 poor and 25 ambiguous responders. For 34 patients, including 9 good, 6 poor, and 19 ambiguous

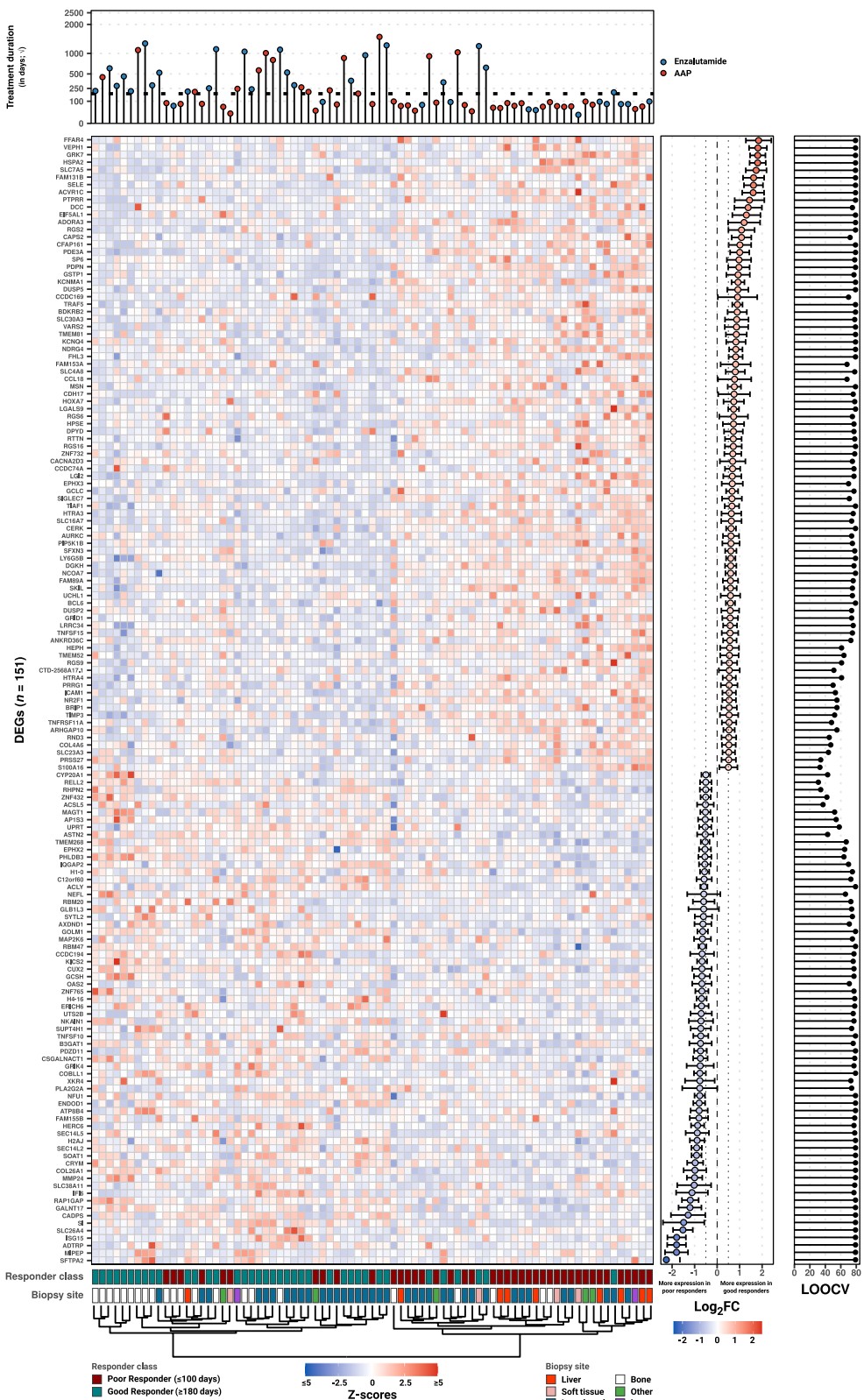

responders, matched WTS was available. This subset was used for the validation of models, that included transcriptomics features (Table 2).

Within our internal validation cohort (n = 76), we correctly predicted 21 out of 28 true good responders and 13 out of 23 true poor responders using the clinicogenomics model, thereby resulting in a sensitivity of 79% and specificity of 57% and an AUC of 0.74. These results are comparable with the results during training. Although

limited by the number of samples during validation, the overall distribution of genomic and clinical features for predicted classes resembles those seen during feature selection (Fig. 6a–n). Survival analysis of the complete internal validation cohort, including the true ambiguous responders, revealed an overall longer ARSI-treatment duration (p = 0.015; log-rank test) for predicted good responders vs. poor responders with respectively a median (and 95%CI) of 187.5 days (143-386) and 116.5 days (90–138) (Fig. 6o).

**Fig. 4 | Differential gene expression analysis between good and poor responders within the discovery cohort reveals 151 putative markers with predictive potential.** Overview of reads counts after variance-stabilizing transformation, shown as Z-scores (negative values in green and positive values in red) for the 151 putative markers with predictive potential between good and poor responders with matching whole-transcriptome data ($n = 47$ and 47, respectively). Each column represents a distinct sample. Samples and genes were clustered (using their respective values as displayed) using Canberra distance and the Ward.D2 method. In descending order, the top and bottom annotation tracks represent: Type and duration of ARSI treatment. Y-axis representing the ARSI treatment duration (in days) whilst the coloring represents the type of given treatment (abiraterone acetate + prednisone in blue, enzalutamide in orange). Y-axis is shown in square-root transformed scale. Responder category. Generalized biopsy location. The inner right-hand boxplot depicts the Log$_2$FC (dot) and Log$_2$FC standard-error (depicted as whiskers) between poor vs. good responders. The outer right-hand boxplot depicts the number of LOOCV-iterations in which each gene found to be statistically significant differentially expressed.

To investigate the confidence of our binary predictions (i.e., predicted poor or good response), we explored whether including a third category of predicted ambiguous responders, capturing uncertain predictions (probability scores of $50 \pm 10\%$), to the clinicogenomics model could result in a better discrimination of poor and good predicted responders (Fig. 6b and p). Using these three prediction categories, similar survival analysis indeed revealed a larger stratification of treatment duration between predicted poor and good responders with a median ARSI-treatment duration (and 95%CI) of 217 days (166–488), 133 days (110–267) and 103 days (84–147) for good, ambiguous and poor predicted responders, respectively, (Fig. 6o) and an increased statistical difference between predicted good and poor responders of $q = 0.0013$ (pairwise log-rank test with BH-correction), compared to the two-group scheme ($p = 0.015$), described above. Although all predicted groups consist of patients from all three true responder categories, only 12% ($n = 3$) of the predicted poor responders is a true good responder, while 23% ($n = 7$) of the predicted good responders is a true poor responder (Fig. 6b–c).

To explore whether the model has additional value in all patient groups, the performance of the clinicogenomics model was tested in uniform pre-treated subgroups within our internal validation cohort (Suppl. Figure 5). In patients, who received 0 or 1 prior therapy, predicted good responders showed a higher median ARSI-treatment duration than predicted poor responders of 266 days (95% CI 172-790, $n = 23$) vs 129.5 days (95% CI 89-NA, $n = 10$), $p = 0.059$. Predicted ambiguous responders showed a median treatment duration of 218 days (95% CI 116-NA) ($n = 9$) in this subgroup (Suppl. Fig. 5a). In patients, who received $\geq 2$ prior therapies, the difference between predicted good and poor responders was less pronounced (median treatment duration (95% CI) 110 days (52-NA) ($n = 7$), 121.5 days (102-NA) ($n = 12$) and 84 days (59–147) ($n = 15$) for good, ambiguous and poor responders, respectively, $p = 0.093$ for good $vs$ poor responders) (Suppl. Fig. 5b). In addition, in patients who did not receive prior enzalutamide, predicted good responders showed a longer treatment duration than predicted poor responders (median 217 days (95% CI 166-488) ($n = 30$) and 90 days (95% CI 84-189) ($n = 15$) respectively, $p = 0.012$). Predicted ambiguous responders showed a moderate median treatment duration of 142 days (95% CI 112-NA) ($n = 15$) (Suppl. Fig. 5c). In patients who did receive prior enzalutamide, no good responders were predicted, while ambiguous and poor predicted responders showed a median treatment duration of 117.5 days (95% CI 63-NA, $n = 6$) and 112 days (95% CI 59-NA, $n = 10$), respectively (Suppl. Fig. 5d). Despite being limited by the sample sizes per subgroup, the relevance of incorporating genomic features was especially visible in patients, who received less prior therapies.

Upon internal validation of the other classification models, especially the clinicotranscriptomics model performed well with a specificity of 50%, sensitivity of 89% and AUC of 0.83. Predicted good responders showed a median treatment duration of 243 days (95% CI 110-NA, $n = 9$), compared to 138 days (95% CI 112-168, $n = 14$) for poor responders ($q = 0.020$) (Table 2, Suppl. Fig. 6).

### External validation cohort
Next, the models were externally validated in the West Coast Dream Team (WCDT) cohort, which included 56 and 77 mCRPC patients for whom WGS and WTS of metastatic biopsies, respectively, was available, and who were treated with ARSI after these biopsies 35. In contrast to the CPCT-02 cohort, clinical outcome in the WCDT cohort was only expressed as overall survival from time of biopsy (OS). Nevertheless, as the correlation between treatment duration and overall survival was clear for patients within the CPCT-02 cohort (mean OS (95% CI) 1613 days (1365–1860) ($n = 66$), 764 days (547–982) ($n = 25$) and 774 days (568–979) ($n = 64$) for true good, ambiguous and poor responders, respectively, $p < 0.001$ for true good vs poor responders), use of the WCDT cohort for external validation was considered justified (Suppl. Fig. 7a).

After application of the clinicogenomics classification model on the external validation cohort, survival analyses of predicted classes revealed overall longer OS ($p = 0.015$; log-rank test) for predicted good responders ($n = 27$) vs. predicted poor responders (n = 29), with a median (and IQR) of 34.1 (25.7-NA) and 17.4 (9.8-31.5) months, respectively, and a hazard ratio (95% CI) of 0.47 (0.28-0.88) (Fig. 7).

In response to the correlation of treatment duration and OS in true responders in the internal cohort and the difference in OS in predicted responders in the external validation cohort, we explored OS in predicted responders in the internal training and validation cohort. Nevertheless, this did not reveal statistically significant differences between predicted poor and good responders ($n = 31$ vs. $n = 36$, $q = 0.88$ and $n = 25$ vs. $n = 30$, $q = 0.15$, respectively; in total 46% of the patients had to be censored for OS) (Suppl. Figure 7b, c).

Finally, we validated the other classification models within the external cohort. As in the internal validation cohort, the clinicotranscriptomics model showed good performance with a hazard ratio of 0.51 (95% CI 0.30–0.86) and a median OS of 1017 days (IQR 771–1691, $n = 42$) and 589 (IQR 487–959, $n = 35$) for predicted good and poor responders, respectively ($p = 0.001$) (Suppl. Fig. 8). Combination of the genomics and transcriptomics in an averaging ensemble model did not result in a better performance than the single models.

### Application of an adapted clinicogenomics model to WES data
To explore the possibility of applying the clinicogenomics model to targeted sequencing data, we assessed the importance of the individual features to the clinicogenomics model. TMB and prior ARSI were more valuable than the number of SVs, deletions and tandem duplications (Suppl Fig. 9). As TMB is also the only feature that could be partially extracted from targeted sequencing data, we developed a simplified model based on TMB and prior ARSI only. Subsequently, we applied this model to WES data, that was extracted from the original WGS data, showing a specificity of 56%, sensitivity of 84% and an AUC of 0.71 in the training cohort and good performance in the internal ($q = 0.001$) and external validation cohort ($p = 0.029$) (Table 2, Fig. 8).

## Discussion
Within this study, we performed an unbiased discovery of biomarkers in whole genomic and transcriptomic data to predict response of mCRPC patients to ARSI. Subsequently, we developed multiple classification models, that can predict response to ARSI in individual mCRPC patients, using machine learning. The clinicogenomics model as well as the clinicotranscriptomics model, both based on prior treatment with ARSI and genomic or transcriptomic features,

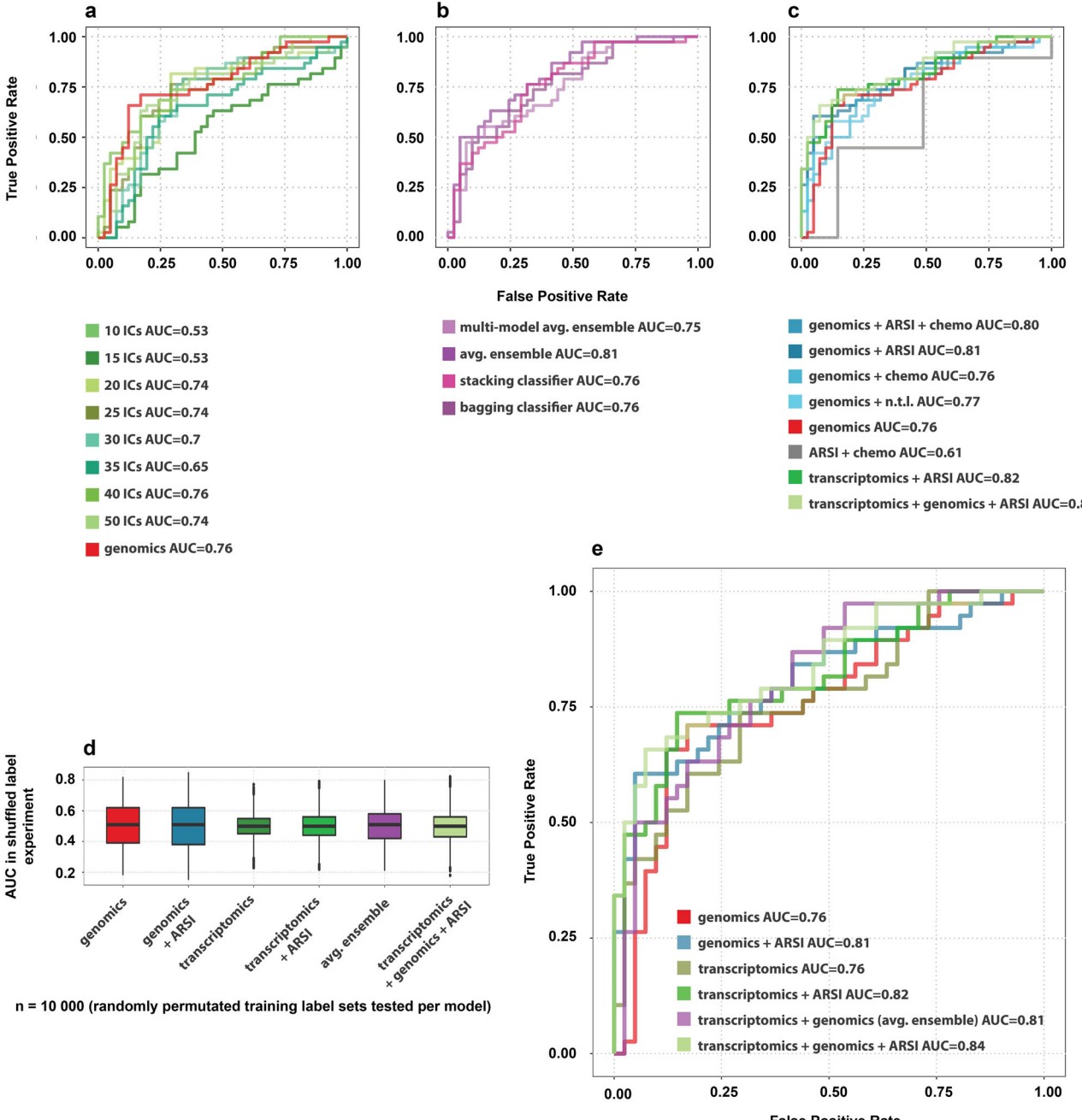

**Fig. 5 | Development of classification models, using Leave-One-Out Cross-Validation (LOOCV) and label permutation. a)**ROC curves with AUC of the genomics-only model and transcriptomics-only models using multiple numbers of independent components (ICs). **b)**ROC curves with AUC of ensemble experiments. Stacking and averaging ensembles were based on genomics-only and best performing transcriptomics (40 ICs) models. Bagging classifier and multi-model averaging ensemble were based on models using randomized subsets of features from combined genomics and transcriptomics data. **c)**ROC curves with AUC of best performing omics modelspo[[, clinical data-only model (ARSI and chemotherapy), and addition of clinical data features to the genomics-only model in multiple combinations. Addition of prior ARSI treatment data to the transcriptomics-only model is also shown. In a final combined experiment, both genomics, transcriptomics and prior ARSI clinical features were jointly tested. N.t.l = number of prior treatment lines. **d)**Shuffled label permutation experiment for the genomics-only, transcriptomics-only, averaging ensemble, clinicogenomics, clinicotranscriptomics and clinicomultiomics (based on addition of prior ARSI) models. Boxplots represent median, inter-quartile range and full range. **e)**Summary of best performing models based on genomics, transcriptomics and clinical data, visualized by ROC curves with corresponding AUC.

respectively, performed well in the training set, internal and external validation cohort. The averaging ensemble model, based on a combination of the genomics and transcriptomics model, performed good as well during training and external validation, but could not distinguish good from poor responders in the internal validation. In addition, we considered it less suitable for clinical application, since it did not outperform the clinicogenomics and clinicotranscriptomics model,

and obtaining the combined sequencing data would be more expensive. The exome-only approximation of the clinicogenomics model showed good results. Application of this model in current clinical practice would be lower in costs than the WGS-based model. In addition, genomics-based models might also be effective with liquid biopsy-obtained sequencing data, which offers possibilities for less invasive response prediction.

**Table 2 | Diagnostic performance of models in training set, internal and external validation**

| Classification model | G | T | C | Training CPCT-02 cohort | | | Internal validation CPCT-02 cohort | | | | | | | | External validation WCDT cohort | | | | Successful models |
|---|---|---|---|---|---|---|---|---|---|---|---|---|---|---|---|---|---|---|---|
| Model components G: Best genomics T: Best transcriptomics (40 ICs) C: Clinical data (prior ARSI) | | | | Matched WGS and WTS: N = 79 True good resp. n = 38 True poor resp. n = 41 | | | WGS-only: N = 76 True good responders n = 28 / True poor responders n = 23 Matched WGS and WTS: N = 34 True good responders n = 9 / True poor responders n = 6 | | | | | | | | Matched WGS and WTS: N = 56 WGS-only n = 56 WTS-only n = 77 | | | | Models with good training performance and significant difference between predicted good and poor responders in all validation cohorts |
| | | | | Spec | Sens | AUC | Spec | Sens | AUC | TD pred. good resp. | TD pred. poor resp. | Log Rank (q) | WGS-only | WGS +/- WTS | HR | OS pred. good resp. | OS pred. poor resp. | Log Rank (p) | |
| | X | X | | 73% | 74% | 0.84 | 50% | 100% | 0.76 | 166.5 (102-790) | 121 (103-NA) | 0.091 | | X | 0.61 (0.33-1.13) | 879 (635-NA) | 581 (466-1123) | 0.110 | |
| | X | X | | 73% | 74% | 0.82 | 50% | 89% | 0.83 | 243 (110-NA) | 138 (112-168) | 0.020 | | X | 0.51 (0.30-0.86) | 1017 (771-1691) | 589 (487-959) | 0.001 | X |
| | X | | | 66% | 76% | 0.81 | 57% | 79% | 0.74 | 217 (166-488) | 103 (84-147) | 0.001 | X | | 0.47 (0.25-0.88) | 1037 (783-NA) | 530 (298-959) | 0.015 | X |
| Aver. ens. | | | | 73% | 68% | 0.81 | 50% | 78% | 0.70 | 133 (110-NA) | 119 (103-NA) | 0.421 | | X | 0.41 (0.22-0.77) | 1168 (635-NA) | 585 (298-959) | 0.005 | |
| | X | | | 49% | 79% | 0.76 | 48% | 71% | 0.63 | 154 (110-439) | 126 (89-189) | 0.435 | X | | 0.48 (0.25-0.91) | 1037 (579-NA) | 606 (469-879) | 0.022 | |
| | | | X | 83% | 58% | 0.76 | 83% | 56% | 0.72 | 154 (102-NA) | 147 (84-NA) | 0.547 | | X | 0.43 (0.25-0.72) | 1017 (782-1691) | 579 (478-959) | 0.001 | |
| Exome-only | X | | | 56% | 84% | 0.71 | 43% | 93% | 0.76 | 199.5 (143-316) | 116.5 (89-138) | 0.001 | X | | 0.5 (0.27-0.94) | 1037 (772-NA) | 474 (296-1123) | 0.029 | X |

Specificity, sensitivity, and Area Under the Curve of classification models in the training set (Leave-One-Out Cross-Validation; LOOCV; n = 79) and internal validation cohort (truly good and poor responders only, binary prediction of good and poor responders; genomics-based models: n = 51, transcriptomics-based model: n = 15). In addition, treatment duration for predicted good and poor responders within the internal validation cohort (based on three-category prediction, including ambiguous predicted responders, genomics-based models n = 76, transcriptomics-based models n = 34, q-value for predicted good vs. poor responders). External validation of the models was performed within the WCDT cohort and shows the hazard ratio (95% confidence interval) for the overall survival of predicted good and poor responders (binary prediction) with the corresponding p value. A summary of the performance of the models is shown in the last column, with successfully and unsuccessfully validated models in green and red, respectively. ARSI androgen receptor signaling inhibitor, AUC area under the curve, HR hazard ratio, ICs independent components, TD treatment duration, OS overall survival, spec specificity, sens sensitivity, WGS whole genome sequencing, WTS whole transcriptome sequencing.

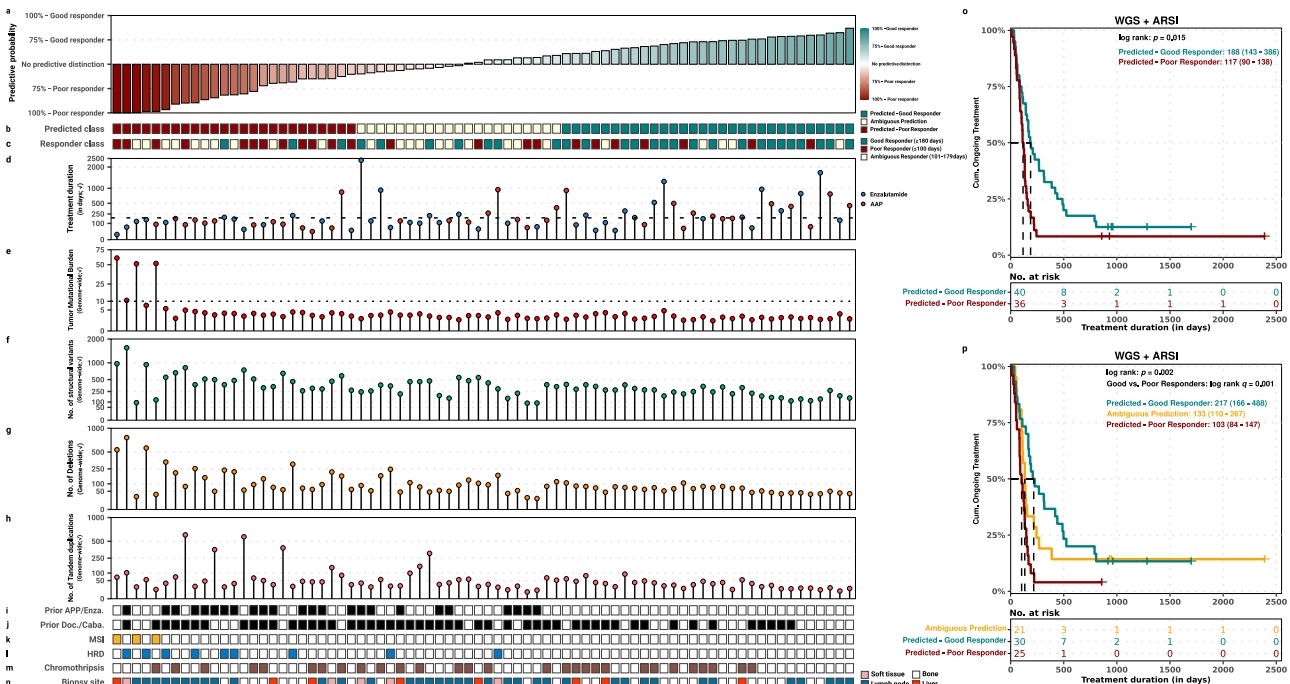

**Fig. 6 | Overview of the clinicogenomics prediction model on the internal validation cohort a-n).** Overview of the prediction scores and genomic features for classifying as a poor or good responder within the internal validation cohort ($n = 76$) using the clinicogenomic classification model. Samples are ordered by descending prediction score (poor to good responder). A prediction score of at least 60% for one of two respective classes (poor or good responder) were required, otherwise samples were designated as ambiguous predictors. For each patient ($n = 76$), the following tracks are shown: **a** Prediction score. The prediction score for either the good or poor responder class. A score of 0.5 means equal likelihood of being either class. **b** Predicted class based on the clinicogenomics classifier. **c** True responder class, based on ARSI treatment duration. **d** Type and duration of ARSI treatment. Y-axis representing the ARSI treatment duration (in days) whilst the coloring represents the type of given treatment (abiraterone acetate + prednisone in blue, enzalutamide in orange). Y-axis is shown in square-root transformed scale. **e** Tumor mutational burden (TMB). The number of genomic mutations averaged per megabase over the entire genome (TMB). Threshold for high-TMB status (TMB ≥ 10) is shown by a horizontal red dotted line. Y-axis is shown in square-root transformed scale. **f** Total no. of structural variants. Y-axis is shown in square-root transformed scale. **g** Total no. of large-scale deletions. Y-axis is shown in square-root transformed scale. **h** Total no. of large-scale tandem duplications. Same as (**g**), but for tandem duplications. **i** Prior abiraterone acetate + prednisone and/or enzalutamide treatment was given to patient. **j** Prior docetaxel and/or cabazitaxel treatment given to patient. **k** Microsatellite-instability (MSI) status. **l** Homologous recombination deficiency (HRD) status as detected by CHORD. **m** Presence of chromothripsis. **n** Generalized biopsy location. **o** Survival analysis using ARSI-treatment duration (in days) and whether patients were still currently receiving ARSI (event) using the two-group scheme of poor and good predictors. Median OS with 95% confidence intervals are shown per strata. **p**)Same as **o**, but between good (≥ 60% probability) poor (≥ 60% probability) and remaining ambiguous predictions.

Clinical differences between true good and poor responders included a lower number of prior treatment lines, less prior treatment with enzalutamide and lower PSA at time of biopsy within good responders, and were expected based on previous studies[36–38]. Prior treatment with enzalutamide could have caused resistance to ARSI within the poor responders, whilst the lower PSA at start of ARSI within the good responders has previously been associated with a better prognosis[36]. However, as baseline PSA was only available for a part of the patients, this could not be included in the training of the classification models[37,38]. Nevertheless, it might be interesting to add this feature to the models during future optimization.

Comparison of genomic characteristics in the internal training set revealed four significantly enriched genomic markers within the true poor vs. good responders. These genomic characteristics included TMB, the total number of structural variants and the total number of tandem duplications and deletions. Genomic aberrations within *AR*, *TP53*, *PTEN*, *RB1*, *CTNNB1,* and chromosomal arms aneuploidies, that were previously associated with ARSI resistance, could not be confirmed within our internal training set[8,15–19,22,23,39]. In addition to genomic characteristics, we observed uniform presence of genes and genesets regulating or attributed to EMT, tumorigenesis, poor survival and/or aggressiveness, having greater expression within the poor responders vs. the good responders. AR-V7, which was previously associated with ARSI resistance, was not differentially expressed in the internal training cohort[8–14].

These discrepancies might be caused by differences in prior therapy and used clinical outcome between the cohorts of the large tissue-based studies (SU2C West Coast and East Coast cohort). In addition, the SU2C West Coast cohort compared enzalutamide-naïve and enzalutamide-resistant patients whilst we investigated pre-treatment biopsies of good and poor responders for WTS analyses. The cfDNA-based studies often studied only a targeted panel of genes and might be confounded by a varying (unknown) tumor fraction within blood.

The CPCT-02 cohort is a diverse cohort of mCRPC patients, who varied in phase of their disease, as is for example illustrated by the wide distribution in number of prior therapies[30]. As we expected treatment duration to be less influenced by disease phase than overall survival, we preferred treatment duration above overall survival as clinical endpoint. Additionally, treatment duration was already available for most patients, while overall survival would have needed to be censored for approximately half of the patients, which would have increased uncertainty in training and internal validation of the model. Nevertheless, treatment duration and overall survival appeared to be highly correlated within the CPCT-02 cohort, justifying its use in the WCDT validation cohort, for which only OS was available[35].

For the clinicogenomics model, additional analyses were performed. The addition of the ambiguous prediction category in the internal validation cohort increased the overall stratification of ARSI-treatment duration for good and poor predicted responders, which

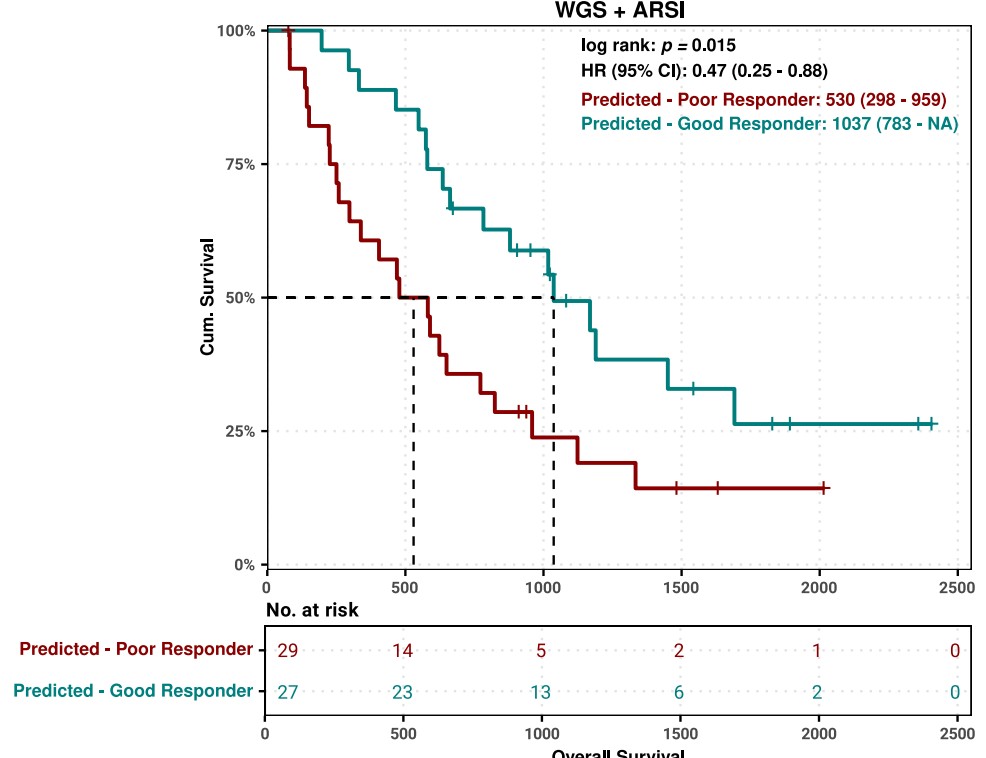

**Fig. 7 | External validation of clinicogenomics model in WCDT cohort.** Overall survival (OS) of patients in the external WCDT cohort from time of first biopsy to death for patients with an ARSI as the next therapy after biopsy. Patients were sub-grouped based on the clinicogenomics model as predicted good or poor responder. Survival curves were visualized using the Kaplan-Meier method and hazard ratios were calculated using Cox proportional hazards regression. *P* value was calculated using the Wald-test/log-rank test. Median OS with 95% confidence intervals are shown per strata.

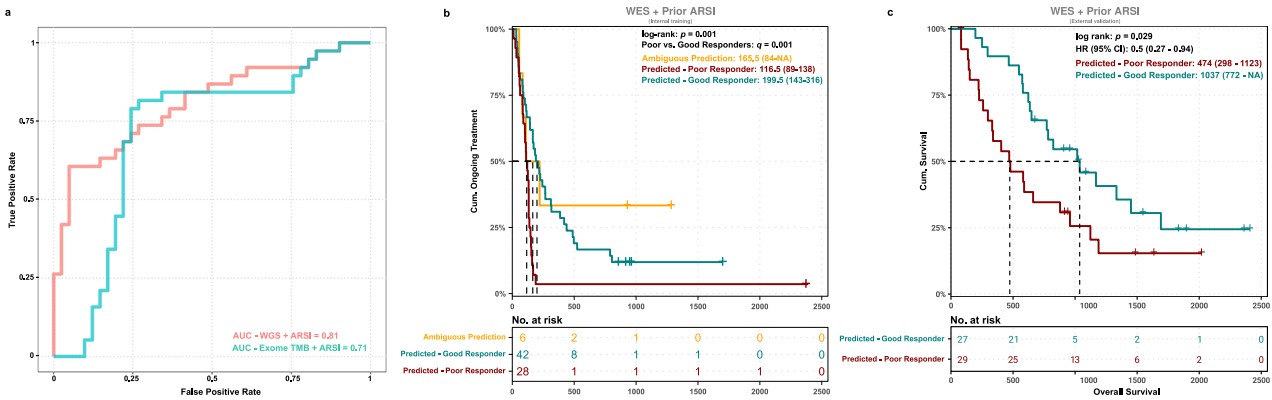

**Fig. 8 | Predictive values of WES-based clinicogenomics model. a** ROC curves and AUCs of WGS clinicogenomics and exome-only (WES) clinicogenomics models in LOOCV. **b** Survival analysis using ARSI-treatment duration (in days) and whether patients were still currently receiving ARSI (event) using the three-group scheme of poor, good and ambiguous predictors for the WES (TMB) + prior ARSI classification model on the internal validation cohort (*n* = 76). Survival curves were visualized using the Kaplan-Meier method and hazard ratios were calculated using Cox proportional hazards regression. Significance was calculated using the Wald-test/log-rank test between all groups (*p*-value) and between poor vs. good responders only (*q*-value). Median OS with 95% confidence intervals are shown per strata. **c** Same as (**b**) for overall survival (OS) of patients in the external WCDT cohort from time of first biopsy to death for patients with an ARSI as the next therapy after biopsy. Patients were sub-grouped based on the WES (TMB) + prior ARSI classification model as predicted good or poor responder.

were predicted with at least 60% predictive probabilities. To explore the additional value of the clinicogenomics model in clinical sub-groups, the performance was tested in uniform pre-treated subgroups within our internal validation cohort. Despite being limited by the number of patients within the subgroups, the relevance of incorpor-ating genomic features was especially visible in patients, that were not heavily pre-treated. Interestingly, this is also the patient group, that would benefit most from improved treatment guidance, as the

number of available therapies is highest in the beginning of the disease course.

By simultaneously interrogating both WGS and WTS with machine learning, we were able to perform an unbiased discovery of biomarkers for response to ARSI in one of the largest cohorts of mCRPC patients with extensive sequencing data. Although clinical studies often have relatively small sample sizes for statistical analyses of whole omics data, machine learning techniques such as dimensionality reduction

with Independent Component Analysis, and LOOCV, enabled the selection of predictive features whilst preventing overfitting. ML-based classification models can, in contrast to the traditional statistical models, determine the most predictive combination of biomarkers from a large set of features and predict response of future individual patients.

Up to now, no biomarkers for response prediction to ARSI are implemented in clinical practice. The most extensively studied biomarker is AR-V7 in CTCs, which presence has been associated with a shorter PFS and OS[14]. However, questions are raised about the confounding prognostic value of AR-V7, and a randomized controlled trial, showing better outcomes for AR-V7 positive patients, who were treated with other therapies than ARSI, hasn't been performed yet[40]. The observed performance of our classification models is not high enough for direct application in the clinical setting. In addition, we can't distinguish whether our models have rather a predictive or prognostic value, as in this study patients were only treated with ARSI. To the best of our knowledge, no other machine learning-based models, that aim to predict response to ARSI in mCRPC patients, have been published.

This study does show the possibilities of response or prognosis prediction based on whole omics data. We also explored the performance of a simplified version of the clinicogenomics model on approximated WES data, which showed good results in the training and validation cohorts as well. Currently, the lower costs of WES are an advantage for the clinical implementation of the simplified model. Nevertheless, the implementation of a wider range of genomic features in the model, as within the original clinicogenomics model, might result in better generalizability in other patient cohorts. Additionally, although whole omics sequencing is not available for all patients nowadays, it is expected that WGS will be more cost-efficient than targeted panel sequencing in the near future due to decreasing costs and increasing number of targeted therapies[41].

In theory, the clinicogenomics model might also be extended to liquid biopsies if sequenced deeply enough to reliably obtain tumor mutational burden and structural variant load. However, obtaining comparably detailed sequencing data from liquid biopsies might be challenging due to the often low tumor fraction and amount of cfDNA, which can be isolated from blood. Nevertheless, analysis on cfDNA does harbor the potential to better capture the inherent heterogeneity of (metastatic) cancer and different clones present throughout the body and would be a worthwhile endeavor to follow up. However, a similar modeling approach to generate a cfDNA-specific model would likely yield better results and might better take into consideration the landscape as seen within cfDNA.

Prediction models could be used to not only stratify patients with a predicted poor response to ARSI for alternative treatments, but also to identify those patients, which are in highest need of additional therapies. In response, clinical trials can focus on the subgroup of patients who respond poorly or moderately to standard-of-care options, such as ARSI, and who would benefit most from the development of additional therapies.

In conclusion, response to ARSI in mCRPC patients can be predicted using machine learning-based classification models, that included whole genomics, transcriptomics and prior treatment data. After optimization and prospective validation, these models could be used to guide treatment decisions and select those patients for clinical trials, that would benefit mostly from the development of therapies.

## Methods
### Study design and patients
With 41 participating hospitals within the Netherlands, the Center for Personalized Cancer Treatment (CPCT) aims to improve cancer treatment by selecting patients for clinical trial participation based on Next Generation Sequencing data of tumor tissue. A list of participating hospitals is available via www.cpct.nl/ziekenhuizen. The prospective

CPCT-02 biopsy study (NCT01855477) has been approved by the medical ethical committee of the University Medical Center Utrecht and has been conducted in accordance with the Declaration of Helsinki. In- and exclusion criteria were published before[20,30,42]. In short, patients were eligible if they had a locally advanced or metastatic solid tumor for which a next line of systemic treatment with a registered anti-cancer agent was indicated, and a safe tumor biopsy could be obtained. All patients provided written informed consent before any study procedures were performed. Compensation for participation was not provided.

For the current analysis, all mCRPC patients, who underwent a successfully sequenced biopsy from a metastatic lesion within the CPCT-02 study between February 2015 and October 2019, and who were subsequently treated with AAP or enzalutamide, were included. As CPCT-02 is an ongoing study with more than 4000 patients, we used a snapshot of the clinical data from December 19th, 2021 for the current analysis (ALEA Clinical). Clinical data collection is performed by trained local data managers and supervised by a central data manager.

### Stratification of patients based on response to ARSI
As the main reason for stop of ARSI is progression of disease and rarely toxicity, patients were stratified according to treatment duration (TD) as surrogate for treatment response[4,5,43]. Patients were stratified in good (TD ≥ 180 days), ambiguous (TD 101–179 days), and poor (TD ≤ 100 days) responders. Cut-off values were based on clinical practice. We considered patients with a treatment duration of ≤100 days as true poor responders, as 100 days (-12 weeks) is typically the first major decision point for treatment (dis)continuation according to the PCWG3 criteria[44]. In addition, another threshold was set at ≥180 days to distinguish the true good responders from the ambiguous responders. To minimize the chance of bias due to incorrectly categorized patients, only the good and poor responder group were used for biomarker discovery and training of the classification model. Nevertheless, for a complete overview of the patient cohort, the ambiguous responders are visualized in the figures and are included during the testing of the classification model.

### Study procedures, sample processing, and sequencing strategies
Study procedures consisted of peripheral blood samples for germline DNA and image-guided core needle biopsies of a metastatic lesion. Biopsies were obtained before start of systemic treatment, independent of line of therapy. Detailed study procedures were published before[30,42]. In short, core needle biopsies were obtained according to standardized protocols and frozen in liquid nitrogen, directly after the procedure. In addition, a tube of blood was drawn. Further sample processing has been performed by the Hartwig Medical Foundation, Amsterdam, the Netherlands. Tumor cellularity was estimated by an experienced pathologist based on a single 6 μm haematoxylin and eosin (H&E) stained section. DNA was isolated from blood and biopsies with ≥30% tumor cellularity, according to supplier's protocol (Qiagen) using the DSP DNA Midi kit and QIAsymphony DSP DNA Mini kit, respectively. Barcoded DNA libraries were prepared from 50–100 ng of genomic DNA (TruSeq Nano LT library preparation, Illumina) and sequenced on HiSeqX generating 2 × 150 read pairs using standard settings (Illumina).

Whole-transcriptome sequencing was performed according to the manufacturer protocols using a minimum of 100 ng total RNA input. Total RNA was extracted using the QIAsymphony RNA kit (QIAGEN, FRITSCH GmbH, Idar-Oberstein, Germany). Paired-end sequencing of (m)RNA was performed on either the Illumina NextSeq 550 platform (2 x 75bp; Illumina, San Diego, CA, USA) and NovaSeq 6000 platform (2 x 150bp; Illumina, San Diego, CA, USA) using manufacturer's protocols.

## Processing and analysis of the whole-genome sequencing data

**Pre-processing of whole-genome sequenced samples.** Whole-genome sequencing samples were pre-processed by the GRIDSS, PURPLE, LINX workflow as detailed previously by Priestley et al. and Cameron et al.[30,45]. PURPLE v3.1, GRIDSS v2.11.1 and LINX v1.16 was used by the Hartwig Medical Foundation (HMF) using a matched-normal design using peripheral blood.

**Additional processing of whole-genome sequenced samples.** From the WGS-data obtained from the HMF, we performed additional processing using a custom workflow as implemented in the R2CPCT (v0.3.2) package. Genomic variants were re-annotated using Variant Effect Predictor[46] (VEP; release 104) based on GRCh37 and GENCODE v38 annotations using the custom workflow available from https://github.com/J0bbie/VariantAnnotation_VEP. In addition, gnomAD[47] (genome and exome v2.1.1) and ClinVar[48] (accessed on 27-09-2021) annotations were added in addition to default VEP annotations.

Genomic (somatic) variants were filtered if they were present in ≥5 samples in the Panel-Of-Normals (PON) of the HMF. In addition, genomic variants were filtered if they were present in the gnomAD exome and/or genome populations with an allele-frequency (AF) of 0.001 and 0.005, respectively. Large structural somatic variants (SV), as detected by GRIDSS (PASS-only), were imported and annotated using the StructuralVariantAnnotation package (v1.10.0) into translocations, deletions, insertions, inversions, tandem duplications and single-breakends (in which the partnering break-end could not be detected).

Genome-wide ploidy, overlapping copy-number segments and their estimated tumor purity-corrected absolute copy-number as derived by PURPLE were used in assessing gene-wise copy-number alterations. If the overlapping copy-number segment of a gene harbored an estimated absolute copy-number ≤0.75, the gene would be classified as an "deep deletion". Similarly, if the estimated absolute copy-number was only half of genome-wide ploidy, it would be classified as an "deletion". If the estimated absolute copy-number was 1.5 times the genome-wide ploidy, it would be classified as an "amplification" and if the estimated absolute copy-number was 3 times the genome-wide ploidy or constituted ≥15 copies, it would be classified as a "deep amplification". For chromosome X and Y, a correction of genome-wide ploidy minus one was used to correct for haploidy in these chromosomes. In addition, if the gene-wise B-allele frequency based on heterozygous germline markers was ≤0.15 or ≥0.85, it would also be classified as a Loss-Of-Heterozygosity (LOH) event. Per gene, this approach was also used to detect deleted or amplified exons using the same criteria.

CHORD (v2.0)[49] was used to assess samples with *BRCA1/BRCA2*-associated homologous repair deficiency using default settings. ShatterSeek (v0.6)[50] was used to detect putative chromothripsis events using best-practice settings as detailed by the authors. The criteria for a chromothripsis-like event were based on the following criteria: (a) total number of intra-chromosomal SVs involved in the event ≥25; (b) max. number of oscillating CN segments (2 states) ≥7 or max. number of oscillating CN segments (3 states) ≥14; (c) total size of chromothripsis event ≥20 megabase pairs (Mbp); (d) satisfying the test of equal distribution of SV types ($p > 0.05$); and (e) satisfying the test of non-random SV distribution within the cluster region or chromosome ($p \leq 0.05$).

## Discovery of genes under evolutionary selection

We performed a dN/dS analysis on somatic mutations (SNV and InDels) using dndscv (v0.0.1.0)[51] on respective genome sequences and transcript annotations using a custom transcript database based on ENSEMBL Genes (v104)/GENCODE (v38) annotations. We performed a dN/dS analysis over the entire discovery cohort ($n = 155$) and on the poor and good responders, separately. Genes-of-interest were selected based on the statistical significance, corrected for multiple hypothesis testing (Benjamini-Hochberg), which integrated all mutation types (missense, nonsense, essential splice-site mutations and InDels; qglobal_cv ≤ 0.1) and/or without InDels (qallsubs_cv ≤ 0.1).

## Unbiased detection recurrent and focal copy-number aberrations and overlap with known drivers

We performed GISTIC2 analysis (v2.0.23) for the WGS-discovery cohort on the PURPLE-derived copy-number segments using tumor purity-corrected absolute copy-numbers as input (log$_2$-transformed − 1, i.e., diploid is set to zero); haploid chromosomes in male samples were corrected by adding a pseudo-count (of 1) prior to log$_2$-transformation. Segments with log$_2$-transformed values ≤10 were set to −10.

GISTIC2 (v2.0.23) was performed using the following settings with default GISTIC2-provided GRCh37 annotations:

gistic2 -b  -seg <segments > -refgene hg19.UCSC.add_-miR.140312.refgene.mat -genegistic 1 -gcm extreme -maxseg 4000 -broad 1 -brlen 0.98 -conf 0.95 -rx 0 -cap 3 -saveseg 0 -armpeel 1 -smallmem 0 -res 0.01 -ta 0.3 -td 0.3 -savedata 0 -savegene 1 -qvt 0.1 -twoside 0

We performed this GISTIC2 analysis for the full discovery cohort ($n = 155$) and separately on the poor and good responder groups.

GISTIC2 output was imported and re-annotated using GENCODE annotations (v38; min. 10 bp overlap) thereby using the wide-peak limits of the recurrent copy-number peaks ($q \leq 0.1$) to classify the region containing the likely target(s) of the recurrent and focal copy-number aberration.

Genes were annotated to GISTIC2 peaks ($q \leq 0.1$) based on the following strategy;

1. All overlapping genes (min. 10 bp) were assigned to the each GISTIC2 peak.
2. If multiple genes overlap a GISTIC2 peak, known driver genes would be used to annotate that peak. E.g., if a GISTIC2 peak overlapped both MYC and a near-adjacent non-driver gene, only MYC would be chosen as possible target.
3. If no overlapping genes could be found, GISTIC2 peaks were annotated with the nearest GENCODE (v38) protein-coding gene.

The peak amplitude thresholds were used to represent the presence (or absence) of the observed GISTIC2 peak within each respective sample; Low amplitude (t > −0.3), Med. amplitude (−0.3 > t > −1.3) and High amplitude (t < −1.3).

## Analysis and quantification of known mutational signatures

Mutational signatures analysis was performed using the MutationalPatterns package (v3.2.0) based on COSMIC signatures (v3.2; single-base substitutions, doublet-substitutions and InDels-based)[52]. Sample-specific signature refitting was done by finding the optimal contribution of the COSMIC signatures (v3.2). Proposed etiologies for the COSMIC signatures were taken from the COSMIC signature database (v3.2).

## Detection of genomic differences between poor and good responders

Differences in genomic characteristics between poor and good responders were tested using a two-sided Mann-Whitney U test with Benjamini-Hochberg correction on the internal validation cohort ($n = 79$). We tested the following genomic characteristics: tumor mutational burden, total number of deletions (SV), total number of inversions (SV), total number of insertions (SV), total number of translocations (SV), total number of tandem duplications (SV), total sum of structural variants per sample and genome-wide ploidy.

Mutual-exclusiveness of mutant genes, chromothripsis status (≥1 chromothripsis event in sample) and HRD-status were assessed between poor and good responders using a two-sided Fisher's Exact

Test with Benjamini–Hochberg correction. Genes with protein-coding mutation(s) and/or deep amplification or deep deletion status were counted as mutants within this analysis.

## Processing and analysis of the whole-transcriptome sequencing data

**Pre-processing of whole-transcriptome data.** Prior to alignment, raw reads (per lane) were pre-processed using fastp (v0.23.2) to trim adapter sequences (paired-end), low-quality bases and perform low-complexity trimming but without a min. length selection on the remainder of the read. Subsequently, these corrected reads are aligned against the human reference genome (GRCh37) with GENCODE (v38)[53] annotations using STAR (v2.7.9a)[54]. Alignment was performed against the full reference genome and also only against the transcriptome to allow for downstream calculation of the fragments per kilo base per million mapped reads (FPKM). Per sample, all lanes (both R1 and R2; paired-end) are used during alignment using the following command:

STAR --genomeDir <GRCh37> --readFilesIn <R1 lanes > <R2 lanes > --readFilesCommand zcat --outFileNamePrefix <prefix> --outSAMtype BAM SortedByCoordinate --outSAMunmapped Within --outSAMattributes All --outFilterMultimapNmax 10 --outFilterMismatchNmax 3 --limitOutSJcollapsed 3000000 --chimSegmentMin 10 --chimOutType WithinBAM SoftClip --chimJunctionOverhangMin 10 --chimSegmentReadGapMax 3 --chimScoreMin 1 --chimScoreDropMax 30 --chimScoreJunctionNonGTAG 0 --chimScoreSeparation 1 --outFilterScoreMinOverLread 0.33 --outFilterMatchNminOverLread 0.33 --outFilterMatchNmin 35 --alignSplicedMateMapLminOverLmate 0.33 --alignSplicedMateMapLmin 35 --alignSJstitchMismatchNmax 5 −1 5 5 --twopassMode Basic --twopass1readsN −1 --runThreadN 10 --limitBAMsortRAM 10000000000 --quantMode TranscriptomeSAM --outSAMattrRGline <sample-specific readgroup>

Post-alignment, duplicate reads were marked using sambamba markdup (v0.8.1)[55] and general alignment metrics (e.g., number of primary-mapped reads) were retrieved using sambamba flagstats (v0.8.1).

**Determining per-gene expression.** Read-counts per gene, from GENCODE annotations (v38), were retrieved using featureCounts (v2.0.3)[56] on primary-aligned reads only with paired-end and strand-specific options:

featureCounts -T 50 -t exon -g gene_id --primary -p -s (1 or 2) -a <GENCODE v38 > -o  <Genome-aligned BAM files>

FeatureCounts was performed on NextSeq 550 WTS samples with -s = 2 whilst NovaSeq 6000 WTS samples were performed with -s set to 1 to address differences in library read-orientations.

Only protein-coding genes were used ($n = 19449$) in all downstream analysis.

**Batch-effect correction.** To remove potential bias regarding site of biopsy, we used the full CPCT-02 mCRPC cohort. We performed differential analysis using DESeq2 per major biopsy site (≥5 samples) versus the rest; the major sites being liver, lymph node, bone and "Other", i.e., liver ($n = 54$) vs. the rest ($n = 267$), lymph node ($n = 159$) vs. the rest ($n = 162$), bone ($n = 72$) vs. the rest ($n = 249$) and "Other" ($n = 36$) vs. the rest ($n = 285$) on all protein-coding genes. Following default DESeq2 analysis (Wald test), we performed LFC-shrinkage using the 'ashr' method[57]. Next, genes with the following criteria were designated as putative biopsy-site (batch-effect) markers: adjusted $p$ ($q$) ≤ 0.05, log$_2$ fold-change standard error (lfcSE) ≤ 1 and |log$_2$ fold-change| ≥ 1. This resulted in a list of 3419 distinct genes, which were significantly enriched (or depleted) within liver, lymph node and/or bone biopsies (Suppl. Table 1). These markers were removed prior to all subsequent whole-transcriptome analysis.

A t-SNE approach ($θ = 0.5$, perplexity = 15, dims = 2, 1000 iterations) was performed to visualize the batch effect of alternate biopsy sites and the effectiveness of the removal within the 155 whole-transcriptome sequenced samples used as discovery cohort within this study and no lingering batch-effects were observed.

**Differential expression analysis between treatment response groups.** We performed differential expression analysis using DESeq2 on all protein-coding genes without the designated biopsy-site specific genes (as described above) between good responders ($n = 38$) vs. poor responders ($n = 41$) within our internal validation cohort. Next, genes with the following criteria were designated as differentially-expressed genes: adjusted $p$ ($q$) ≤ 0.05, an average read count over all samples (baseMean) ≥ 25, Log2FC standard error ≤ 1 and |log2FC| ≥ 0.5.

**Quantification of AR-V7 expression.** We quantified the percent spliced in (PSI) of AR-V7 by comparing the number of junction-reads which spanned AR exon 2 and cryptic exon 3 (AR$_{V7}$) vs. the number of junction-reads spanning AR exon 1 and exon 2 (AR$_{e12}$) and dividing them appropriately:

$$\text{PSI}_{\text{ARV7}} = \frac{\text{AR}_{V7} \text{ reads}}{(\text{AR}_{V7} \text{ reads} + \text{AR}_{e2312} \text{ reads})} \quad (1)$$

## Design of ML-classification models for prediction of response to ARSI

The internal cohort of patients with matched WGS and WTS was divided in a training- and internal validation set of 70% ($n = 79$) and 30% ($n = 34$) of the samples, respectively. Good and poor responders were randomly divided. Ambiguous responders were included in the validation set only. The training set was used in LOOCV ('LeaveOneOut' from *sklearn.model_selection*) to determine model performance and then the full training set was used to train a model, that was applied on the internal validation set. To perform validation on the external cohort, the same training set was used to train a classifier, but for WTS data, additional preprocessing steps were applied (see below). Figure 2 shows the machine learning model design and evaluation steps (figure made in BioRender.com).

Additional experiments with hyperparameter tuning in grid search were applied for the best performing models in LOOCV. The hyperparameter combinations were evaluated based on accuracy score ('sklearn.metrics.accuracy_score').

**Classification input preprocessing and model design.** The four significantly divergent genomic features between good and poor responders to ARSI, namely TMB, total number of structural variants, total number of tandem duplications and total number of deletions, were centered and scaled prior to classification. Standard scaling was a necessary pre-processing step based on the comparison of genomic feature distributions in the training, internal and external cohorts (Suppl. Figure 10). To train the genomics and genomics-clinical covariate models, Logistic Regression classifier was applied ('LogisticRegression' from *sklearn.linear_model, solver = 'liblinear'*).

The raw transcriptomics data was TMM transformed (*edgeR*) and centered and scaled ('StandardScaler' from *sklearn.preprocessing, with_mean = True, with_std = True*). To perform dimensionality reduction, sparse PCA[58], conventional PCA[59] and Independent Component Analysis[60] were evaluated. While evaluating the cumulative explained variance of the principal components is a widely used approach to select the optimal number of components to describe the dataset with PCA, this information is not available when applying sparse PCA and Independent Component Analysis (additionally the latter being an entirely different approach). Therefore, to compare sparse PCA ('sparsePCA' from *sklearn.decompostion*), PCA ('PCA' from *sklearn.decomposition*) and Independent Component Analysis ('FastICA' from *sklearn.decomposition*), these models were first applied on the training dataset with target sparse component number ('n_components')

ranging from 10 to 50. Afterwards, a Linear Support Vector Classifier (Linear SVC) ('LinearSVC' from *sklearn*, *penalty = 'l2'*, *loss = 'squared_hinge'*, *C = 1.0*, max_iter = 10'000) was trained on a given set of components which was subsequently calibrated ('CalibratedClassifierCV' from *sklearn*). Lastly, all dimensionality reduction-based classification models were evaluated based on AUC (Suppl. Fig. 11) and the best performing model was chosen for internal and external validation.

**Combining genomic and transcriptomic models and data.** As an attempt to exploit the strength of both models and data types, different ensembling techniques were applied. A stacking classifier was built using 'StackingClassifier' from *sklearn.ensemble* where the prediction probability output of both models was used in final_estimator=*LogisticRegression()* to calculate the final prediction. The averaging ensemble approach was carried out by averaging the prediction probabilities from the best transcriptomic model (40 independent components or ICs) and the genomic model. A multi-model averaging ensemble was built by averaging predictions from transcriptomics-genomics averaging ensemble model pairs from 100 randomized evaluations. In each individual averaging ensemble, the transcriptomics model used *n* randomly selected components (*max. n* = 40; from the best performing 40 independent components (ICs) based transcriptomics data decomposition) from which the prediction was averaged with the genomics model. Each of these individual ensemble results were then aggregated and averaged over the 100 experiments. Lastly, a bagging classifier was built using 'BaggingClassifier' from *sklearn.ensemble* with *n_estimators* = 100, *max_samples* = 1.0, *bootstrap* = True, *oob_score* = True and *max_features* = 0.35, by randomly sampling from a joint set of transcriptomics (40 ICs) and genomics features for each individual learner. Boosting ensemble methods that require extensive subsampling were not assessed due to the limited size of the training set (79 samples). Moreover, the main goal of ensembling was to reduce variance (due to potential overfitting) and not bias. All tested ensemble approaches were evaluated in LOOCV in the initial model design step.

**Addition of prior treatment data to classification models.** Additionally, the WGS-only and WTS-only classification models were extended with baseline clinical variables: AAP/enzalutamide pretreatment, chemotherapy pretreatment and the number of treatment lines. The former two clinical variables were binary (0 – not received, 1 – received) while the number of treatment lines was ranging from 0 to 9. The clinicogenomics, clinicotranscriptomics and a joint WTS + WGS + ARSI models were trained and evaluated in LOOCV.

**Shuffled labels experiments on final classification models.** Shuffled label experiments were carried out on the best performing WTS-only, WTS + ARSI, WTS + WGS + ARSI, WGS-only, WGS + ARSI and ensemble (WTS + WGS) models. By permuting the sample labels, the corresponding distribution of the null hypothesis (='there is no meaningful feature pattern that can be used to distinguish between poor and good responders') can be estimated. The label shuffling procedure measures how likely it is that the observed classifier accuracy can be obtained by chance. For each classification model, the sample labels were randomly shuffled in 10'000 iterations using *numpy.random.shuffle()*. Afterwards, LOOCV was performed in each iteration, where the shuffled labels were used in model training and then a left-out test sample label was predicted in each fold. The same shuffled labels were used for all classification models in each iteration.

### Validation of classification models in internal cohort
For the validation of the WGS-only and clinicogenomics classification models, the genomic features were centered and scaled within the internal cohort data. To perform dimensionality reduction on the

transcriptomics data of the internal validation cohort, the best independent components model that was already fitted on the training cohort was applied to transform the dataset. Following the dimensionality reduction, the transformed data was used not only in the WTS-only model but in the combined WTS + ARSI, WTS + WGS + ARSI and ensemble (WTS + ARSI) models to predict responder groups in the internal validation cohort.

Diagnostic accuracy and predictive values were evaluated for the true good and poor responders within the internal validation cohort, for comparison with the training set, as well as within the complete internal validation cohort, including true ambiguous responders too. Additionally, treatment duration and overall survival were compared in the predicted groups. Subgroup analyses in similarly pre-treated patients were performed for the clinicogenomics model to evaluate the additional predictive value of genomics to clinical data.

### Validation of classification models in external cohort
Classification models were validated in the external West Coast Dream Team cohort (WCDT), which included mCRPC patients treated with ARSI after biopsy[21,35]. WGS was available for 56 patients, while WTS was available for 77 patients. Clinical outcome was defined as overall survival from time of biopsy to death of any cause.

Data from the WCDT cohort was pre-processed by applying the same steps as in the CPCT-02 cohort. The genomic features were centered and scaled prior to prediction. The transcriptomics data was TMM-normalized, then centered and scaled. To account for batch-effect and general inter-domain variability of the CPCT-02 and the WCDT transcriptomics datasets, a domain adaptation method (PRECISE) was used[61]. PRECISE first employs Independent Component Analysis separately on the two transcriptomics matrices. Then using the independent component datasets, it infers a so-called consensus representation. First, the consensus representation was fitted on the training data (70% of CPCT) and calculated with 'ConsensusRepresentation' with *n_factors* = 40, *n_pv* = 40, *dim_reduction* = *ica*, *n_representations* = 40, *mean_center* = True and *std_unit* = True. Then, both the training set and the external validation set were transformed and a Linear SVC classifier (with the same parameters as described in '*Classification input preprocessing and model design*') was trained on the transformed training data. The combined WTS + ARSI, WTS + WGS + ARSI and ensemble (WTS + ARSI) models were also re-trained on the transformed training data. Afterwards, the external validation cohort sample labels were predicted with each model as good or poor responder and OS in both groups was compared.

As the utilized transcript annotations differed between the internal training set and the external cohort (GENCODE v38 and GENCODE v28), missing genes were filtered out from the internal training set. To evaluate the potential effect of the missing genes on the classification, we compared a classifier that was trained on the full transcriptomics dataset and a classifier that was trained on the filtered transcriptomics dataset in the initial LOOCV step (Suppl. Fig. 12).

### Application of an adapted clinicogenomics model to WES data
Importance of the features in the clinicogenomics model was assessed in LOOCV. Individual importance values were determined based on the fitted Logistic Regression model coefficients, which were accessible from the trained model (*model.coef*[0]). Unbiased interpretation of these coefficients required that all the input features were scaled prior to model training (see Suppl. Fig. 10 and Classification input preprocessing and model design). The obtained importance values were visualized in a bar plot, with error bars indicating the standard deviation of values across all LOOCV folds (Suppl. fig. 9). To determine the potential of a prediction model based on tumor mutation burden as could be observed with WES (coupled with information on prior ARSI), we subsampled somatic mutations found within our WGS to include those found within exonic regions only (GENCODE v38). TMB was

recalculated (using total Mb of exonic regions rather than the entire genome) to generate a WES proxy for TMB.

A Logistic Regression model was trained on the approximated WES feature (exome TMB) and prior ARSI treatment information in LOOCV, then the performance was assessed based on AUC (Fig. 8a). Internal and external validations were carried out in the same fashion as for the WGS-based clinicogenomics model (see Fig. 8b, c).

## Statistical designs

Clinical characteristics of good and poor responders were compared using the appropriate statistical test, based on number of variables and normality distribution (t-test, non-parametric test or Fisher's Exact test). *P*-values were adjusted for multiple testing using the Bonferroni method. Unless otherwise stated, statistical tests were performed in a two-sided manner. Treatment duration and OS in the predicted groups were visualized in Kaplan Meier curves. Good and poor responders were compared using log rank tests. Statistical tests were performed in IBM SPSS Statistics (v28.0.1.0 (142) and the statistical platform R (v4.1.1)).

Genomic differences (i.e., TMB, total SV and total deletions, translocations, insertions, inversions and tandem duplications and genome-wide ploidy) were tested using the two-sided Wilcoxon Rank-Sum Test with multiple testing correction (Benjamini-Hochberg). Statistical tests were performed in the statistical platform R (v4.1.1). For visualization, *p*-values (or *q*-values) are visualized as $*(p < 0.05)$, $**(p < 0.01)$ and $***(p < 0.001)$.

## Reporting summary

Further information on research design is available in the Nature Portfolio Reporting Summary linked to this article.

## Data availability

The WGS, RNA-Seq and corresponding clinical data used in this study was made available by the Hartwig Medical Foundation (Dutch non-profit biobank organization) after signing a license agreement stating data cannot be made publicly available via third party organizations. Therefore, the data are available under restricted access and can be requested upon by contacting the Hartwig Medical Foundation (https://www.hartwigmedicalfoundation.nl/applying-for-data/) under the accession code: DR-071[20]. In addition, we performed analysis on the patients of the previously reported WCDT cohort, who were treated directly after biopsy with ARSI and for who WGS and/or WTS was previously performed[21,35]. For a detailed description of data availability, we refer to this paper[21]. Requests for data can be directed towards prof. dr. Felix Feng, E: Felix.Feng@ucsf.edu. The data generated in this study are provided in the Source Data file. Source data are provided with this paper.

## Code availability

The initial workflows and software for the processing of the WGS data are available at https://github.com/hartwigmedical/. Any additional custom code and scripts used within this study (processing, analysis, and visualization) have been deposited on Zenodo: DOI: 10.5281/zenodo.7712610[62] The custom R-based workflow (R2CPCT) used to further analyze the WGS-data obtained from HMF and CPCT-02 study is available on GitHub under the GPL-3.0 license: https://github.com/J0bbie/R2CPCT The code used to further annotate genomic variants (as retrieved from HMF) using VEP is available on GitHub under the GPL-3.0 license: https://github.com/J0bbie/VariantAnnotation_VEP.

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

## Acknowledgements

This research was financially supported with an unrestricted grant by Johnson & Johnson (ML; 212082PCR3014) and Astellas Pharma (ML; Lolkema/NL-72-RG-11). In addition, we would like to acknowledge the Erasmus MC Cancer Computational Biology Center (CCBC) and Hartwig Medical Foundation (HMF) for sharing their expertise and computational resources.

## Author contributions

Ad.J., A.D., and J.V.R. wrote the manuscript, which all authors critically reviewed. Ad.J. managed clinical data assessment, which was supervised by Rd.W. and M.L. A.D., supervised by Jd.R and Jv.R. performed the bioinformatics analyses. M.L. is PI of the CPCT-02 study. F.F. provided the external validation cohort, for which M.S. performed the validation analyses.

## Competing interests

RdW has speaker/advisory roles at Sanofi, Bayer, and Astellas, advisory roles at Orion, Hengrui, and Merck US, and received institutional research grants from Sanofi and Bayer. FF received personal fees as a consultant for Janssen, Myovant, Roivant, Novartis, Astellas, Foundation Medicine and Exact Sciences, as a member of the scientific advisory board of Bayer, SerImmune, Bristol Meyers Squibb (BMS), Bluestar Genomics, Blue Earth Diagnostics, and Tempus, as an advisor with stock options at Artera, and as former co-founder with ownership interests of PFS Genomics (relationship termed in April 2021. JdR is co-founder of Cyclomics BV. ML received advisory role/speaker fees from Incyte, Amgen, Janssen Cilag B.V., Bayer, Servier, Roche, Pfizer, Sanofi Aventis Netherlands BV, and Astellas, and has received institutional research funding from Sanofi, JnJ, Merck, and Astellas. All other authors declare no competing interests.
