## [Peer Review File · Nature Communications]

Predicting response to enzalutamide and abiraterone in metastatic prostate cancer using whole-omics machine learningREVIEWER COMMENTS

Reviewer #1 (Remarks to the Author): clinical expertise in prostate cancer biomarkers

In the manuscript titled "Predicting response to androgen receptor signaling inhibitors in metastatic castration resistant prostate cancer patients through machine learning-based analysis of whole genome and transcriptome sequencing data" Anouk de Jong et al. integrate clinical, genomic and transcriptomic data to predict clinical outcomes in men with metastatic prostate cancer. To this end, the authors leverage well annotated datasets, define clinical, genomic and transcriptomic features that associate with outcome and build a combined classifier which achieves an AUC of 0.81. External validity of the model is then further confirmed in a separate cohort. Overall, the manuscript answers a potentially critical clinical question. The strength of manuscript lies in 1. the use of well-annotated cohorts, 2. the application of state-of-the-art machine learning approaches, 3. a very clear and concise data presentation. However, there are several concerns regarding the broader applicability and impact of this study.

1. Although the authors demonstrate that an integrated machine learning approach can augment outcome predictions, the clinical relevance of this finding and its potential implementation are not established. Given that most patients (even in academic centers) do not undergo WGS and WTS, it would be of great practical relevance to study the model's performance with targeted panel sequencing data as input (ideally in a separate cohort).

2. The authors state that their model predicts response to ARSI therapy. However, from the data presented in the manuscript it is not clear how treatment response to this specific therapeutic intervention can be differentiated from overall prognosis (i.e. is the model predicting aggressive disease rather than features that are specific to ARSI response).

3. As the authors have noted, several other reports have shown that genomic alteration in AR, RB1 and TP53 are prognostic and predictive in mCRPC, raising questions about potential differences in cohort composition between the European training cohort (used in this study) and the US-based cohorts that have been reported previously. This issue would need to be addressed.

4. Further validation in the SU2C international dream team cohort would strengthen the external validity of the study.

Reviewer #2 (Remarks to the Author): expert in machine learning

Jong et al presented their analysis on the prediction of androgen receptor signalling inhibitors (ARSI) in metastatic castration-resistant prostate cancer (mCRPC) patients using logistic regression on genomics, clinical and transcriptomics data of selected, global features. The findings are interesting, and the methods applied are rooted in well-tested methodology. However, the reviewer thinks that the presentation of data/results need to be much improved, the rationale explained, and additional analyses are needed to strengthen the conclusion of the paper.

1. Data presentation

a. First of all, the reviewer would like to point out that Figure legend texts are too small to be readable (across all Figures!) Figure resolution is poor and thus makes it challenging to judge the results. The reviewer believes these are shown (albeit very unclear especially in the low-res heatmap) but still led to the co-reviewer raising the following:

i. It might be useful to show the distribution of the different variables (mostly continuous ones), which would probably shed more light on the choice of scaling methods for these variables.
ii. Exploration of how the validation cohort distribution and variables might have differed from the training cohort, and the concordance algorithms (besides transcriptomics) for the datasets.

b. Also, The authors did not replicate AR, TP53, PTEN, RB1, CTNNB1 mutations to be associated with treatment response; the statement linked to Fig S1e, but in that figure only chr aneuploidies were shown

2. Methodology & analyses

Overall, the choice of methods especially in ML is not well-explained, as raised by the co-reviewer:

a. Use of sparse PCA as a feature reduction method is well-established but not without its difficulties. Hence covariance of the features, captured variance of features for the number of

principal components used and the loss of information might be explored. There are various other feature selection and regularisation algorithms that might be more useful for this dataset, and might lead to better results.

b. The paper gives a comparison of various machine learning methods used on the dataset for combining transcriptomics and genomics data. However, more details about these experiments of combination of these features are missing. Transcriptomics and genomics data being a little different and with selective covariances between them might need more careful selection of features as well as ensembling. The reviewer would like to learn about these experiments of the combination of these variables.

c. It would be interesting to know more about the choice of algorithms in this paper. Why was stacking classifier ensemble method chosen over other ensemble methods? Why is logistic regression chosen as a method of choice?

d. This paper reports sensitivity vs specificity in the results. Was there any hyperparameter optimization done? If yes, which parameters were optimised? Was specificity favoured compared to sensitivity? It will be interesting to see what the AUC after these experiments.

e. Were there also experiments based on WTS+clinical data, or a combination of all the data? What were the findings?

3. Results

a. The results mention overfitting without dimensionality reduction. It would be interesting to see the same results without PCA on the validation cohort too. Also, in the reviewer's knowledge, most ML models using mRNA-Seq data at the gene level (without dimensionality reduction) typically perform in previous ML studies on Omics dataset so once that's tried if there's discrepancies may need to be explained.

b. More details on the overlap of the models, not only in terms of numbers of true positives or negatives but the similarity of features for the patients selected or the dissimilarities between them might give a more detailed insight into the working of these models as well as might help in ensembling techniques chosen.

c. Details of the feature importances across the different trained models and discussion of them in comparison to prior work would add more value to this manuscript.

Minor:

1. The limitation of liquid-biopsy may be overstated (in theory if analysed to high depth these may be possible, "However, liquid biopsy-based analyses are mostly targeted to a certain set of genes or provide only superficial information, like whole genome copy numbers, and rely on patients having a high tumor-derived cfDNA fraction in the blood. Therefore, liquid biopsies are less suitable for the discovery of potential novel biomarkers, that predict outcome to treatment."

2. Main Workflow Fig, the arrow makes it look like transcriptomics are only avail for poor responders

Reviewer #3 (Remarks to the Author): expertise in prostate cancer genomics

Metastatic castration resistant prostate cancer (mCRPC) patients show variable response to androgen receptor (AR) inhibitors like abiraterone or enzalutamide. The authors developed a machine learning-based prediction model to identify patient groups who would respond to AR inhibitors treatment. mCRPC patients treated with abiraterone acetate + prednisone (AAP) and enzalutamide from CPCT-02 were used as discovery cohort. Whole genomics (WGS) and whole-transcriptomics (WTS) were obtained from biopsies before treatment, in combination with clinical variables, the prediction model successfully stratify patients into good and poor responders towards AAP and enzalutamide. This model was further validated in an external cohort WCDT.

Overall, I think this is a very interesting study that constructed a novel model to predict patient outcome based on pre-treatment biopsies. However, I do have some questions from which I believe the paper can benefit by addressing them.

Major concerns:

1. The treatment duration was used for classification of good or bad response instead of overall survival or progression free survival. In line 346, the authors mentioned that this was to minimize the effect of prior therapy. But it seems that prior therapy was integrated into the clinicogenomics model, which was used for WCDT cohort validation. More justifications are needed for choosing treatment duration instead of overall survival.

Also, in Supfig.7a, the authors showed that overall survival had high correlation with treatment duration. But predicted responder groups based on treatment duration showed no significant stratification, especially in training group (Supfig.7b). Was the stratification defined by clinicogenomics model? If so, what are the potential causes of this results?

2. The applicability of the developed clinicogenomics model was not emphasized enough. How is the performance of clinicogenomics model compared to standard diagnosis or other machine learning-based model? Is the model strong enough for other larger patient cohorts? Are there any possibilities to extend this model from genomic data in solid biopsies to less-invasive liquid biopsies?

3. The authors proposed that clinicogenomics model was based on 4 genomics features and clinical information. More explanations are needed to justify choosing the 4 features over the others. Also, the clinical information needs further clarification. Is prior AR inhibitor treatment the sole factor integrated into the model or there are other clinical variables?

Minor concerns

1. The authors built WGS-only and WTS-only model and merged them into one classification model. Would there be any difference to combine the matched WGS and WTS data and identify novel features to build a new classification model?

2. From the clinicogenomics model, are there any novel biomarkers or signature pathways that can be identified as diagnostic markers?

3. The classification of ambiguous responders and good responders need further justification. In line 342-344, the authors mentioned that adding ambiguous prediction category in the internal validation cohort increased the overall stratification. What would be if we add ambiguous group into the training category? Would it benefit the predictive probabilities?

4. Biopsies were obtained from metastatic sites. Are there any prostate tumor biopsies sequencing data available before treatment?

RESPONSE REVIEWER COMMENTS

We thank the reviewers for their useful and constructive comments. We answered all reviewer's questions and remarks with special attention for comparison of our method with targeted sequencing, comparison with other diagnosis methods and inclusion of further details of the machine learning methodology.

Reviewer #1 (Remarks to the Author): clinical expertise in prostate cancer biomarkers

In the manuscript titled "Predicting response to androgen receptor signaling inhibitors in metastatic castration resistant prostate cancer patients through machine learning-based analysis of whole genome and transcriptome sequencing data" Anouk de Jong et al. integrate clinical, genomic and transcriptomic data to predict clinical outcomes in men with metastatic prostate cancer. To this end, the authors leverage well annotated datasets, define clinical, genomic and transcriptomic features that associate with outcome and build a combined classifier which achieves an AUC of 0.81. External validity of the model is then further confirmed in a separate cohort. Overall, the manuscript answers a potentially critical clinical question. The strength of the manuscript lies in 1. the use of well-annotated cohorts, 2. the application of state-of-the-art machine learning approaches, 3. a very clear and concise data presentation. However, there are several concerns regarding the broader applicability and impact of this study.

1. Although the authors demonstrate that an integrated machine learning approach can augment outcome predictions, the clinical relevance of this finding and its potential implementation are not established. Given that most patients (even in academic centers) do not undergo WGS and WTS, it would be of great practical relevance to study the model's performance with targeted panel sequencing data as input (ideally in a separate cohort).

*For model development, we preferred WGS and WTS to enable unbiased discovery of potential new biomarkers. This resulted in genome-wide characteristics as best-predicting biomarkers. These can typically not be extracted from targeted panel sequencing. However, we assessed the individual importance of the significant features to the performance of the clinicogenomics model in LOOCV (See **Methods and Supplemental Figure 9**) and found that prior ARSI and TMB were most important.*

In an attempt to answer this reviewer's question, we approximated targeted panel sequencing data by extracting whole exome sequencing (WES) data from the WGS data of the CPCT cohort (approximately 3% of the complete data set). Subsequently, we tested the model performance with those features that were available using WES data (TMB of the exome and prior ARSI). This resulted in an AUC of 0.71, specificity of 56% and sensitivity of 84% in comparison to an AUC of 0.81, specificity of 66% and sensitivity of 76% of the original

clinicogenomics model in LOOCV (Figure 8, Table 2). Subsequently, the model was validated on the WGS-only internal validation cohort and achieved an AUC of 0.76, specificity of 43% and sensitivity of 93% in comparison to an AUC of 0.74, specificity of 57% and sensitivity of 79% of the original clinicogenomics model (Table 2). The exome TMB and prior ARSI-based model was also validated on the external cohort and showed good performance with a HR of 0.5 (95% CI 0.27-0.94), $p = 0.029$ for predicted good vs. poor responders (Figure 8).

Thus, both the original clinicogenomics model based on WGS and the new model based on WES performed well in all cohorts. Advantage of the WES-based model would be the lower costs, although costs for WGS and WTS are decreasing and are expected to be more cost-efficient than targeted panel sequencing in assessing a myriad of genetic alterations in the near future. In addition, implementation of a variety of features in the model, as within the original model, might result in better generalizability in other patient cohorts.

The above described information has been added to the Methods, Results and Discussion section of the manuscript.

2. The authors state that their model predicts response to ARSI therapy. However, from the data presented in the manuscript it is not clear how treatment response to this specific therapeutic intervention can be differentiated from overall prognosis (i.e. is the model predicting aggressive disease rather than features that are specific to ARSI response).

We agree with the reviewer that prognostic and predictive value can't be distinguished from each other based on data of ARSI-treated patients only. We have added some text discussing this to the Discussion section.

3. As the authors have noted, several other reports have shown that genomic alteration in AR, RB1 and TP53 are prognostic and predictive in mCRPC, raising questions about potential differences in cohort composition between the European training cohort (used in this study) and the US-based cohorts that have been reported previously. This issue would need to be addressed.

Within the SU2C East Coast cohort (Abida et al, Genomic correlates of clinical outcome in advanced prostate cancer, PNAS, 2019), overall survival of 128 and ARSI treatment duration of 108 taxane-naive, first-line ARSI patients are compared to genomic alterations (WES and WTS). Within this cohort, only RB1 alteration was significantly associated with poor survival, whereas alterations in RB1, AR, and TP53 were associated with shorter time on treatment with an ARSI. Interestingly, AR-V7 expression was not significantly correlated to time on treatment with an ARSI, as in the CPCT-02 cohort.

Within the SU2C West Coast cohort (Chen et al, Genomic Drivers of Poor Prognosis and Enzalutamide Resistance in Metastatic Castration-resistant Prostate Cancer, European Urology, 2019), overall survival from time of mCRPC diagnosis of 101 patients is compared to WGS and WTS data. This cohort consists of patients with and without prior ARSI treatment. Within this cohort, the presence of two RB1 DNA alterations was independently associated with poor OS among men with mCRPC. Upon comparison of enzalutamide-naive and enzalutamide-resistance patients, the Wnt/ β -catenin pathway played an important role in enzalutamide resistance, with differential pathway expression of the Wnt/ β -catenin pathway and enrichment of β -catenin mutations in enzalutamide-resistant patients were found.

Differences between the SU2C cohorts and the CPCT-02 cohort are the median number and type of prior therapy (first-line vs variation in number of prior lines, taxane-naive vs taxane-resistant) and used clinical outcome (overall survival from time of mCRPC diagnosis vs ARSI treatment duration). In addition, there are differences in the sequencing resolution (WES vs WGS) and employed analysis workflows to detect somatic mutations, copy-number alterations and structural variants which may explain minor technical differences in observed mutational rates between both cohorts but would not reflect in different number of detected somatic mutations within AR, RB1 and/or TP53. In addition, SU2C compared enzalutamide-naive and enzalutamide-resistant patients whilst we investigated pre-treatment biopsies of good and poor responders for WTS).

In conclusion, the data on the prognostic value of these genes has been based on different patient populations using various methods. We were unable to confirm these findings, therefore larger efforts and meta-analyses using compounded data sets would be the best method to verify their true value. The discussion of the manuscript has been adapted accordingly.

4. Further validation in the SU2C international dream team cohort would strengthen the external validity of the study.

We thank the reviewer for the suggestion. We indeed aimed to validate our model in the SU2C East Coast cohort as well. However, for this cohort, only WES and WTS data is available. Thus, validation of the genomics-based models was not possible within this cohort. To ensure fair comparison of all models, we decided to use the SU2C West Coast cohort as the only external validation cohort.

Reviewer #2 (Remarks to the Author): expert in machine learning

Jong et al presented their analysis on the prediction of androgen receptor signaling inhibitors (ARSI) in metastatic castration-resistant prostate cancer (mCRPC) patients using logistic regression on genomics, clinical and transcriptomics data of selected, global features. The findings are interesting, and the methods applied are rooted in well-tested methodology. However, the reviewer thinks that the presentation of data/results need to be much improved, the rationale explained, and additional analyses are needed to strengthen the conclusion of the paper.

1. Data presentation

a. First of all, the reviewer would like to point out that Figure legend texts are too small to be readable (across all Figures!) Figure resolution is poor and thus makes it challenging to judge the results. The reviewer believes these are shown (albeit very unclearly especially in the low-res heatmap) but still led to the co-reviewer raising the following:

We thank the reviewer for this remark. Full resolution figures were provided separately from the manuscript file during submission. We will ask the editor to provide the full resolution images to the reviewer.

i. It might be useful to show the distribution of the different variables (mostly continuous ones), which would probably shed more light on the choice of scaling methods for these variables.

*We thank the reviewer for this suggestion. Plots showing the distribution for different variables with and without scaling in the training cohort have been added to the Supplemental material (**Supplementary Fig 10**).*

ii. Exploration of how the validation cohort distribution and variables might have differed from the training cohort, and the concordance algorithms (besides transcriptomics) for the datasets.

*Plots showing the distribution for different variables with and without scaling in the external validation cohort have been added to the Supplemental material (**Supplementary Fig 10**).*

b. Also, The authors did not replicate AR, TP53, PTEN, RB1, CTNNB1 mutations to be associated with treatment response; the statement linked to Fig S1e, but in that figure only chr aneuploidies were shown

*We had previously tested these genes for enrichment or depletion (Fisher's Exact Test and Benjamini-Hochberg correction, as described in the manuscript) but did not find significant differences between poor and good responders for AR, TP53, PTEN, RB1 and CTNNB1. We have now also correctly added these results into **Fig. S1f**, we apologize for the previous omission.*

2. Methodology & analyses

Overall, the choice of methods especially in ML is not well-explained, as raised by the co-reviewer:

a. Use of sparse PCA as a feature reduction method is well-established but not without its difficulties. Hence covariance of the features, captured variance of features for the number of principal components used and the loss of information might be explored. There are various other feature selection and regularisation algorithms that might be more useful for this dataset, and might lead to better results.

We used sparse PCA as a means to dimensionality reduction of the high-dimensional transcriptomics data. While in non-sparse, conventional PCA the obtained loadings are orthonormal, in sparse PCA, the orthonormality does not hold for the loadings due to the applied regularization constraint. Due to this, it is not straightforward to calculate the explained variance of the components. In our work, we used the sparse PCA implementation from scikit-learn, where the orthogonality is not enforced and there is no option to extract the explained variance of the components. Therefore, the captured variance of the sparse principal components cannot be directly explained in the manuscript. If requested, the authors can provide further reference that supports this claim.

*To further address the reviewer's question, we tried other dimensionality reduction algorithms, namely independent component analysis (ICA) and conventional PCA. Multiple classifiers were trained on the components extracted with these methods and their performance was compared to the sparse PCA-based transcriptomics model (**Supplementary Figure 11**). Furthermore, the explained variance of principal components used in the PCA-based models were visualized (see figure below).*

*The best performance was reached by a model trained on 40 independent components (ICs) with an AUC of 0.76. Consequently, the best performing transcriptomics model in LOOCV (40 ICs components) was tested on the internal and external validation sets. On the internal cohort, this classifier reached an AUC of 0.72, with a specificity of 83% and a sensitivity of 56%. Moreover, the model was also validated on the external cohort where it performed well, with the predicted good and poor responders showing a significant difference in OS ($p = 0.001$)(**Table 2, Supplemental Figure 6 and 8**).*

Differential expression analysis (DESeq2) was already explored as a means to feature selection conventionally performed in mRNA-seq datasets. The findings and related conclusions are described in the Results section of the manuscript.

Figure: Cumulative and individual explained variance up to 50 components using PCA on the transcriptomics data matrix.

b. The paper gives a comparison of various machine learning methods used on the dataset for combining transcriptomics and genomics data. However, more details about these experiments of combination of these features are missing. Transcriptomics and genomics data being a little different and with selective covariances between them might need more careful selection of features as well as ensembling. The reviewer would like to learn about these experiments of the combination of these variables.

*The ROCs of all the main combined experiments are shown on **Figure 5**. Moreover, we extended the ensemble experiments and these additional experiments are now described in the Methods (**see also Figure 5b**). Both the bagging classifier and the multi-model averaging ensemble approaches in the extended analysis (see Methods) involve randomized feature selection as a means of testing different subsets of combined transcriptomics and genomics features.*

c. It would be interesting to know more about the choice of algorithms in this paper. Why was the stacking classifier ensemble method chosen over other ensemble methods? Why is logistic regression chosen as a method of choice?

*We would like to point out that besides the stacking classifier, averaging ensemble was also applied, albeit based on only two classifiers: the best performing transcriptomics model and the genomics model. In order to address this question, we further explored multiple ensemble types. The updated **Figure 5b** shows the ROC of these ensemble experiments. The best performing ensemble type remained the averaging ensemble, which was based on the best transcriptomics and the genomics models, with an AUC of 0.81 in LOOCV. However, within the internal validation an AUC of 0.70 and no significant difference between treatment*

duration of predicted good and poor responders ($p = 0.421$) was found, while the model performed well in the external validation (**Table 2, Supplemental Figure 6 and 8**).

For both the genomics and transcriptomics-based models, multiple classifier types were assessed but only the best performing model types were included in the manuscript and the model validations on internal and external cohorts. The best performing model type for transcriptomics data was primarily Linear Support Vector Classifier (LinearSVC) and for the genomics data was Logistic regression. Regarding the choice of the tested classifiers, we were motivated by the need for interpretability, which is an important aspect for clinical applications. Secondly, due to the notably small size of the training set (79 samples), we refrained from using complex, non-linear classification models to avoid potential overfitting.

d. This paper reports sensitivity vs specificity in the results. Was there any hyperparameter optimization done? If yes, which parameters were optimised? Was specificity favoured compared to sensitivity? It will be interesting to see what the AUC after these experiments.

Grid search of hyperparameters was applied in the LOOCV experiments for the genomics, clinicogenomics and best transcriptomics models. The optimization did not yield performance improvement over the parameter settings described in Methods [results not shown]. The evaluation of hyperparameter value selection was based on accuracy score ('sklearn.metrics.accuracy_score'), therefore specificity was not favored over sensitivity, nor the hyperparameter selection process was influenced by these metrics.

We now included a short summary of these experiments in the Methods section of the revised manuscript.

e. Were there also experiments based on WTS+clinical data, or a combination of all the data? What were the findings?

We added these experiments to the manuscript, using the updated best transcriptomics model (based on 40 independent components). **Table 2** now contains the AUCs, specificity and sensitivity of all the aforementioned experiments: best transcriptomics model + prior ARSI and best transcriptomics model + prior ARSI + genomics features. The ROCs of these experiments were added to **Figure 5b-c**. Both of these models were validated on the internal and external cohort as well (**Supplemental Figure 6 and 8**).

3. Results

a. The results mention overfitting without dimensionality reduction. It would be interesting to see the same results without PCA on the validation cohort too. Also, in the reviewer's knowledge, most ML models using mRNA-Seq data at the gene level (without dimensionality reduction) typically perform in

previous ML studies on Omics dataset so once that's tried if there's discrepancies may need to be explained.

We would like to thank the reviewer for this interesting suggestion. A raw transcriptomics data based model was trained using all the individual features (16,665 genes) and the 79 training samples. This model reached an AUC of 0.58 in LOOCV and was further validated on the external cohort. Subsequent survival analysis revealed that this model performed poorly on the external cohort, without significant difference in OS between the predicted poor and good responder groups ($p = 0.27$ log-rank test) [see Kaplan-Meier plot below]. Therefore, overfitting of the classifier on the training dataset without dimensionality reduction was confirmed.

b. More details on the overlap of the models, not only in terms of numbers of true positives or negatives but the similarity of features for the patients selected or the dissimilarities between them might give a more detailed insight into the working of these models as well as might help in ensembling techniques chosen.

Figure 3 shows a detailed landscape of different genomics features while **Figure 4** shows the result of the initial feature discovery on the WTS data across the full CPCT-02 cohort. **Figure 6** shows the genomic feature values in respect to predicted responder groups in the

internal cohort. **Supplementary Figure 9** shows the feature importance values in the genomics model, while the interpretation of these values and how they influence prediction is described in **Methods**. Additionally, ensemble experiments with randomized feature selection were evaluated using transcriptomics and genomics features (see **Methods, Figure 5b**)

c. Details of the feature importances across the different trained models and discussion of them in comparison to prior work would add more value to this manuscript.

*We thank the reviewer for this suggestion. **Supplementary Figure 9** was added to the manuscript which shows the coefficients of clinicogenomics model. To the best of our knowledge, there are no other machine learning-based models that aim to predict response to ARSI in mCRPC patients. Therefore no direct comparison to prior machine learning methods was made.*

Minor:

1. The limitation of liquid-biopsy may be overstated (in theory if analysed to high depth these may be possible, “However, liquid biopsy-based analyses are mostly targeted to a certain set of genes or provide only superficial information, like whole genome copy numbers, and rely on patients having a high tumor-derived cfDNA fraction in the blood. Therefore, liquid biopsies are less suitable for the discovery of potential novel biomarkers, that predict outcome to treatment.”

We understand the comment of the reviewer and removed the statement of superficial information from the manuscript.

2. Main Workflow Fig, the arrow makes it look like transcriptomics are only avail for poor responders

We thank the reviewer for the comment and adapted the figure.

Reviewer #3 (Remarks to the Author): expertise in prostate cancer genomics

Metastatic castration resistant prostate cancer (mCRPC) patients show variable response to androgen receptor (AR) inhibitors like abiraterone or enzalutamide. The authors developed a machine learning-based prediction model to identify patient groups who would respond to AR inhibitors treatment. mCRPC patients treated with abiraterone acetate + prednisone (AAP) and enzalutamide from CPCT-02 were used as discovery cohort. Whole genomics (WGS) and whole-transcriptomics (WTS) were obtained from biopsies before treatment, in combination with clinical variables, the prediction model successfully stratify patients into good and poor responders towards AAP and enzalutamide. This model was further validated in an external cohort WCDT. Overall, I think this is a very interesting study that constructed a novel model to predict patient outcome based on pre-treatment biopsies. However, I do have some questions from which I believe the paper can benefit by addressing them.

Major concerns:

1. The treatment duration was used for classification of good or bad response instead of overall survival or progression free survival. In line 346, the authors mentioned that this was to minimize the effect of prior therapy. But it seems that prior therapy was integrated into the clinicogenomics model, which was used for WCDT cohort validation. More justifications are needed for choosing treatment duration instead of overall survival.

We thank the reviewer for the comment. Within the CPCT-02 cohort, progression free survival was not defined. As ARSI is in general discontinued due to progression of disease and not due to adverse events, we used treatment duration as a surrogate for progression free survival. We considered overall survival less suitable for model development, since we expected that overall survival would be more biased by phase of the disease, including number of prior treatments, performance status, tumor load etc., compared to treatment duration, especially because patients in the CPCT-02 cohort varied widely in phase of their disease. In addition, overall survival data was only yet available for approximately half of the patients. Many patients would've been censored, which would have resulted in increased uncertainty in training and validation of the model.

After we decided to use treatment duration as clinical outcome, we still integrated prior therapy into the clinicogenomics model, as this was a significant difference between the defined good and poor responder group and thus, was expected to optimize model performance.

The manuscript has been clarified accordingly (see Discussion).

Also, in Supfig.7a, the authors showed that overall survival had high correlation with treatment duration. But predicted responder groups based on treatment duration showed no significant stratification, especially in training group (Supfig.7b). Was the stratification defined by clinicogenomics model? If so, what are the potential causes of this results?

In Supplemental Figure 7b and c, the stratification was indeed defined by the clinicogenomics model. This model has been trained to predict treatment duration and not overall survival. Although treatment duration is correlated to both overall survival and predictions based on the clinicogenomics model, these correlations are not one to one. Thus, when predictions based on the clinicogenomics model are correlated to overall survival, it is expected that this relationship will be less strong. In addition, the confidence of overall survival data in the CPCT-02 cohort has been limited by the high number of censored patients (n = 71, 46%).

2. The applicability of the developed clinicogenomics model was not emphasized enough.

How is the performance of clinicogenomics model compared to standard diagnosis or other machine learning-based model? Is the model strong enough for other larger patient cohorts? Are there any possibilities to extend this model from genomic data in solid biopsies to less-invasive liquid biopsies?

No biomarkers for response prediction to ARSI are yet implemented in clinical practice.

The most extensively studied biomarker for response prediction to ARSI is AR-V7 in CTCs. Within the PROPHECY study, two AR-V7 assays were both associated with a shorter PFS and OS, after adjusting for CTC count and clinical prognostic factors. However, questions are raised about the confounding prognostic value of AR-V7, and a randomized controlled trial, showing better outcomes for AR-V7 positive patients, who were treated with other therapies than ARSI, hasn't been performed yet. Therefore, it's advised to not use the AR-V7 assay outside clinical trials yet.

To the best of our knowledge, there are no other machine learning-based models that aim to predict response to ARSI in mCRPC patients.

Our models would be strong enough for other larger patient cohorts. However, up to now, no larger patient cohorts with available WGS data were published.

In theory, the clinicogenomics model might be extended to liquid biopsies if sequenced deeply enough to reliably obtain tumor mutational burden and structural variant load. However, to obtain comparably detailed sequencing data from liquid biopsies might be challenging due to the often low tumor fraction and amount of cfDNA which can be isolated from blood. Analysis on cfDNA does harbor the potential to better capture the inherent heterogeneity of

(metastatic) cancer and different clones present throughout the body and would be a worthwhile endeavor to follow up. However, a similar modeling approach to generate a cfDNA-specific model would likely yield better results and might better take into consideration the landscape as seen within cfDNA.

Unfortunately, we currently do not have matched cfDNA from these patients to pursue this further but would like to thank the reviewer for this interesting thought.

The comments, described above, were added to the Discussion section of the manuscript.

3. The authors proposed that clinicogenomics model was based on 4 genomics features and clinical information. More explanations are needed to justify choosing the 4 features over the others. Also, the clinical information needs further clarification. Is prior AR inhibitor treatment the sole factor integrated into the model or there are other clinical variables?

*Based on differences between good and poor responders in the baseline table (**Table 1**), clinical features were integrated (as described in Results section, paragraph 'Addition of clinical data to the WGS- and WTS-based classification models'). We tested the additional value of adding multiple combinations of prior treatment data (**Fig 5c**) to the genomics model and addition of prior ARSI resulted in the highest increase in performance. Therefore only prior information on treatment with ARSI was implemented in the final models. Other clinical data was not implemented as these did not show a significant difference between good and poor responders in the baseline table, except for PSA at baseline. However, as PSA at baseline was only available for a minority of the patients, it was not possible to train the models on this feature. In future studies, it would be interesting to explore the predictive value of baseline characteristics in more detail (as described in the second paragraph of the Discussion).*

*In addition, the choice of our four genomic features (TMB, total structural variant load (SV) and total no. of large-scale deletions and no. of large tandem duplications) was based on their statistically significant differences observed within our training dataset (70% of CPCT-02) between poor and good responders (**Fig. 1Sa-d**). We also tested the absolute numbers of large-scale insertions, large-scale interchromosomal translocations, large-scale inversions and genome-wide ploidy but did not observe statistically significant differences for these features (data not shown).*

As these are readily-available features with WGS and may reflect underlying somatic processes, we opted to use these genomic features for further investigation within our genomics and clinicogenomics models.

Minor concerns

1. The authors built WGS-only and WTS-only model and merged them into one classification model. Would there be any difference to combine the matched WGS and WTS data and identify novel features to build a new classification model?

*The reviewer is suggesting an early integration strategy (as opposed to the late integration strategy). This was explored in the now extended ensemble experiments (see Methods) where subsets of combined transcriptomics and genomics features were used to build a bagging classifier and a multi-model averaging ensemble. The results of these experiments in LOOCV are shown in **Figure 5b** and described in the Results and Methods section of the manuscript.*

Another option for early integration would have been to perform dimensionality reduction on the joint WGS and WTS datasets to discover new features and to subsequently apply this feature set in a classification model. However, the problem of this approach is that, due to the sheer number of variables contained in the WTS data compared to the genomics features, the WTS data would dominate the dimensionality reduction process and the obtained components. Therefore, we did not explore this approach.

2. From the clinicogenomics model, are there any novel biomarkers or signature pathways that can be identified as diagnostic markers?

*As described within our manuscript, we performed a differential expression analysis and gene-set enrichment on the initial good vs. poor responders within the training dataset. This indeed revealed 151 differentially-expressed genes (see Methods; **Fig. 4**) and several perturbed gene-sets (**Fig. S2a**) but did not point toward any uniform or singular novel mechanism; rather a plethora of (more aggressive) mechanisms being at play within the poor responders such as epithelial-mesenchymal transition (EMT), TNF- α signaling, TGF- β signaling and Hippo-Merlin signaling, among others. With our updated WTS analysis, we do however observe that genes involved with androgen response are different between good and poor responders and have added this to the manuscript (line 164).*

We had not performed a similar analysis of the validation set (after classification with the clinicogenomics model) as the number of samples with matching WTS (N=34) was deemed as likely too few to produce significant and uniform results which would not have already been identified within the training dataset (N=79).

3. The classification of ambiguous responders and good responders need further justification. In line 342-344, the authors mentioned that adding ambiguous prediction category in the internal validation cohort increased the overall stratification. What would be if we add ambiguous group into the training category? Would it benefit the predictive probabilities?

Model development was based on a binary distribution of clinical outcome (good vs. poor responders) in order to emphasize any potential differences between outer ends of ARSI treatment durations. The ambiguous category consists of patients without a distinct clinical response to ARSI and is therefore a group that might introduce a lot of noise in the training of a multinomial model (poor, ambiguous and good response).

Therefore, we considered that addition of a third category would rather lead to a more complicated and noisier model and we preferred to train a binary model on a set of patients with a more distinct clinical outcome (as described in the Methods, paragraph 'Stratification of patients based on response to ARSI').

4. Biopsies were obtained from metastatic sites. Are there any prostate tumor biopsies sequencing data available before treatment?

*Unfortunately, there is no matching sequencing data available of primary tumor biopsies at diagnosis within the CPCT-02 cohort for patients who also underwent a metastatic biopsy. Several distinct biopsies of the primary tumors before the start of ARSI (n = 13) were excluded from the analyses (**Figure 1**). We preferred biopsies from metastatic sites, as these represent the newest and most aggressive part of the disease. However, the genetic heterogeneity, that is often seen in primary tumors, is indeed missed by using biopsies from metastatic sites. A combination of our study and liquid biopsies might solve this issue, though liquid biopsies are less suitable for deeply sequenced whole-omics analyses due to the often limited amount of ctDNA obtained.*

REVIEWERS' COMMENTS

Reviewer #1 (Remarks to the Author):

The authors have addressed the majority of my comments.

Reviewer #2 (Remarks to the Author):

The authors have addressed my comments.

Reviewer #3 (Remarks to the Author):

The authors have sufficiently addressed my questions and concerns. The quality of the manuscript has been improved and I have no further question.

RESPONSE TO REVIEWERS' COMMENTS

We thank the reviewers for their previous useful and constructive comments and are delighted that they agree with our revised version of the manuscript. No scientific changes were performed during the editorial revisions.